



# Spring melt pond fraction in the Canadian Arctic Archipelago predicted from RADARSAT-2

Stephen E.L. Howell[1], Randall K. Scharien[2], Jack Landy[3] and Mike Brady[1]

[1]Climate Research Division, Environment and Climate Change Canada, Toronto, M3H 5T4, Canada
[2]Department of Geography, University of Victoria, Victoria, V8W 2Y2, Canada
[3]School of Geographical Sciences, University of Bristol, Bristol, BS8 1QU, United Kingdom

*Correspondence to*: Stephen E.L. Howell (Stephen.Howell@Canada.ca)

**Abstract.** Melt ponds form on the surface of Arctic sea ice during spring, influencing how much solar radiation is absorbed into the sea ice-ocean system, which in turn impacts the ablation of sea ice during the melt season. Accordingly, melt pond fraction ($f_p$) has been shown to be a useful predictor of sea ice area during the summer months. Sea ice dynamic and thermodynamic processes operating within the narrow channels and inlets of the Canadian Arctic Archipelago (CAA) during the summer months are difficult for model simulations to accurately resolve. Additional information on $f_p$ variability in advance of the melt season within the CAA could help constrain model simulations and/or provide useful information in advance of the shipping season. Here, we use RADARSAT-2 imagery to predict and analyze peak spring $f_p$ and evaluate its utility to provide predictive information with respect to sea ice area during the melt season within the CAA from 2009-2018. The temporal variability of RADARSAT-2 $f_p$ over the 10-year record was found to be strongly linked to the variability of mean April multi-year ice area and the spatial distribution of RADARSAT-2 $f_p$ was found to be in excellent agreement with the sea ice stage of development prior to the melt season. RADARSAT-2 $f_p$ values were in good agreement with the peak $f_p$ observed from *in situ* observations but were found to be ~0.05 larger compared to peak MODIS $f_p$ observations. Statistically significant detrended correlations between RADARSAT-2 $f_p$ and summer sea ice area were found for several regions within the CAA. Our results show that RADARSAT-2 $f_p$ can be used to provide predictive information about summer sea ice area for a key shipping region of the Northwest Passage.

## 1 Introduction

Arctic sea ice extent during the summer months has declined considerably over the satellite record (Serreze et al., 2007; Stroeve et al., 2012; Peng and Meier, 2017). Surface melt ponds, which form on sea ice during the spring, play an important role in the decay of sea ice and seasonal reduction in ice extent because they influence how much solar radiation is absorbed into the sea ice-ocean system (Eicken et al., 2004). Specifically, the accumulation of meltwater on the surface of the sea ice lowers the albedo from ~0.8 to between 0.2-0.4 and enhances melt (Perovich et al., 2002). The topographical constraints over multi-year ice (MYI) imposed by hummocks typically result in MYI exhibiting a lower melt pond fraction



($f_p$) compared to seasonal first-year ice (FYI) (Grenfell and Perovich, 2004; Polashenski et al., 2012; Landy et al., 2015). With Arctic sea ice transitioning from a MYI to FYI dominated icescape (Maslanik et el., 2011), the lower $f_p$ of MYI will

gradually be replaced with the higher $f_p$ of FYI, facilitating even more sea ice energy absorption and further enhancing sea ice melt (Perovich and Polashenski, 2012).

Predicting the state of Arctic sea ice several months in advance is challenging and recently, the sea ice prediction community has focused efforts on the development and utilization of dynamical forecast models (e.g. Chevallier et al., 2013; Sigmond et al., 2013; Guemas et al., 2016). Despite these recent efforts, rapidly changing Arctic sea ice conditions will

continue to necessitate improved sea ice forecasting capabilities (Eicken, 2013). Accordingly, prognostic $f_p$ schemes have been integrated in climate models and have shown to exert a strong influence on summer sea ice area and extent (Flocco et al., 2010; Flocco et al., 2012). Schröder et al. (2014) found a strong correlation between model-simulated May $f_p$ and the observed September sea ice extent. Observed $f_p$ has also demonstrated significant predictive skill for September ice extent from late-July onwards (Liu et al., 2015). However, while $f_p$ estimates for the entire Arctic can be provided by model

simulations, more representative and higher spatial resolution observational estimates at regional and pan-Arctic scales are much more difficult to obtain.

Optical remote sensing is the most widely utilized approach to estimate large-scale $f_p$ from space (e.g. Markus et al., 2003; Tschudi et al., 2008; Rösel et al., 2012; Istomina et al., 2014; Webster et al., 2015; Lee et al., 2020) but cloud cover remains a significant problem. Techniques for retrieving $f_p$ using advanced quad-polarization and compact-polarization mode

synthetic aperture radar (SAR) imagery, at C- and X-band frequencies, have also been developed (Scharien et al., 2014; Fors et al., 2017; Li et al., 2017) but they are limited in systematic spatial application because the required polarization modes are not always available from wide-swath imagery. However, using the winter backscatter from widely available Sentinel-1 SAR imagery, Scharien et al. (2017) recently demonstrated a technique for predicting spring $f_p$ over the entire Canadian Arctic Archipelago (CAA) 3-4 months in advance of melt pond formation. These $f_p$ predictions have potential utility in

seasonal summer sea ice area and extent forecasts as early as April.

The CAA is a collection of islands located in Northern Canada (Figure 1) whose waterways are sea ice covered between fall and spring. It is an active region for marine shipping and has recently experienced an increase in summer shipping activity (Pizzolato et al., 2014). Model simulations have been utilized to understand the current and predicted future variability of sea ice conditions in the CAA (e.g. Dumas et al., 2006; Sou and Flato, 2009, Howell et al., 2016; Laliberté et

al., 2016; Hu et al., 2018; Laliberté et al., 2018) but it still remains challenging because complex sea ice dynamic and thermodynamic processes are often not accurately resolved in its narrow channels and inlets. In addition, the response of the CAA to climatic change is perhaps counter-intuitive as longer melt seasons are resulting in increased MYI import from the Arctic Ocean during the summer months (Howell and Brady, 2019). Since $f_p$ is linked to summer sea ice melt processes (e.g. Eicken et al., 2004; Skyllingstad and Polashenski, 2018) additional information on $f_p$ variability within the CAA could

improve our understanding of regional summer melt processes, help constrain model simulations and facilitate safer shipping activity in upcoming years.





In this study, we extend the work of Scharien et al. (2017) and investigate predicted $f_p$ variability within the CAA over the longer-term record available from RADARSAT-2. Specifically, (i) we estimate the predicted seasonal peak $f_p$ in the CAA using RADARSAT-2, (ii) evaluate the spatiotemporal variability of $f_p$ in the CAA from 2009-2018 (iii) compare RADARSAT-2 $f_p$ values to Sentinel-1 $f_p$ values from Scharien et al. (2017), *in situ* $f_p$ observations from Landy et al. (2014) and Moderate Resolution Image Spectroradiometer (MODIS) $f_p$ values from Rösel et al. (2012) and (iv) investigate the utility of RADARSAT-2 $f_p$ to provide predictive information about sea ice area in the CAA during the summer melt season.

## 2 Methodology

### 2.1 Data

The primary dataset used in this analysis was 5.405 GHz (wavelength, λ = 5.5 cm; C-band) SAR imagery in ScanSAR wide mode at HH polarization from RADARSAT-2 acquired over the CAA (Figure 1) in April from 2009-2018 (Table 1). RADARSAT-2 ScanSAR wide mode imagery has a spatial resolution of 100 m with an incidence angle range of 20.0° to 49.3°.

*In situ* observations of melt pond fraction on landfast FYI were obtained in two consecutive years from sites in the CAA using a terrestrial Light Detection and Ranging (LiDAR) system (Landy et al., 2014) (Figure 1, green star). In 2011, the site was located in Allen Bay on FYI with relatively rough surface topography, whereas in 2012, the site was located in Resolute Passage on FYI with relatively smooth topography. At each site, a time-series of $f_p$ observations were collected within the same 100 x 200 m area of the ice over a 2 to 3 week period following melt onset, covering three of the four stages of melt pond evolution detailed in Eicken et al. (2004). The LiDAR system produces dense measurements over snow or sea ice with specular reflection over melt ponds allowing melt pond fractions to be retrieved with an accuracy better than 5% (Landy et al., 2014). These observations allow us to evaluate how well predicted $f_p$ from RADARSAT-2 resolve the peak $f_p$ of seasonally-evolving sea ice coverage.

We also made use of 8-day composite satellite observations of $f_p$ obtained from the MODIS Arctic melt pond cover fractions dataset for the period of 2009-2011 (Rösel et al., 2012) and weekly sea ice area and stage of development observations obtained from the Canadian Ice Service Digital Archive (CISDA) regional ice charts for the period of 2009-2018 (Tivy et al., 2011).

### 2.2 Estimating $f_p$ from RADARSAT-2

RADARSAT-2 $f_p$ was determined using a modified approach to that described by Scharien et al. (2017). Their approach determines the second stage of the seasonal melt pond evolution cycle when $f_p$ is at its peak (Eicken et al., 2003; Polashenski et al., 2012) using Sentinel-1 Extra Wide (EW) swath imagery obtained during April in within the CAA. April corresponds to late winter sea ice conditions in the CAA, when sea ice growth has reached its maximum and spring warming has yet to begin. Their approach was developed by relating the winter period HH gamma nought ($\gamma^{\circ}$) backscatter in decibel (dB) from Sentinel-1 to peak $f_p$ observations in 1.7 m spatial resolution GeoEye-1 imagery, from spatially coincident image



segments that represented homogeneous FYI and MYI regions. The result was that $\gamma°$ can be converted to $f_p$ using the following equation:

$$f_p = -0.221 - 0.041(\gamma°) \qquad (1)$$

In equation (1), $\gamma°$ was found to explain 73% of the variability in $f_p$ (Scharien et al., 2017).

105       In this study, all the available HH polarization RADARSAT-2 imagery over the CAA in April from 2009-2018 (Table 1) were first calibrated to $\gamma°$ which minimizes the influence of incidence angle more so than with sigma nought ($\sigma°$) (Small, 2011). RADARSAT-2 images were then speckle filtered using a 5x5 Lee Filter and spatially registered to a common map projection. Finally, $\gamma°$ was converted to $f_p$ by applying Equation (1) to each RADARSAT-2 image. For each year, the corresponding RADARSAT-2 $f_p$ images in April were mosaicked together to cover the entire spatial domain of the CAA.

Constructing a mosaic over a large region such as the CAA presents certain challenges with SAR imagery, particularly incidence angle variability. Even with the use of $\gamma°$, Scharien et al. (2017) found that because of varying incidence angles associated with different ScanSAR images that $f_p$ striping can still occur within the CAA in the mosaicked image. Our approach here was to average out incidence angle variability by taking advantage of large amount of overlapping RADARSAT-2 imagery within the CAA (i.e. 90 to 159 images; Table 1) together with the fact that the majority of the sea

ice in the CAA is landfast (immobile) during April which results in a temporally stable $f_p$ for all April images. To produce a RADARSAT-2 $f_p$ mosaic within the CAA for each year, we calculated the mean $f_p$ for each overlapping pixel using all of each year's RADARSAT-2 April images that effectively helped to reduce $f_p$ striping across the CAA.

       The root-mean square error (RMSE) of $f_p$ based on equation (1) is 0.085 (Scharien et al., 2017). While calculating the mean $f_p$ of the overlapping image pixels helps reduce striping across the CAA, it also adds additional uncertainty and its

effectiveness depends on the number of overlaps. In order to quantify the additional uncertainty ($RMSE_{R2}$), we used the mean and maximum standard deviation of RADARSAT-2 $f_p$ of all pixels within the CAA calculated from 2009-2018 ($f_{std}$) together with a range of pixel overlaps (n) in the following equation:

$$RMSE_{R2} = [(f_{std}/n^{0.5})^2 + 0.085^2]^{0.5} \qquad (2)$$

Since RADARSAT-2 imagery is acquired operationally, overlapping images vary interannually but pixel overlaps across the

CAA were typically between 6-12. Figure 2 illustrates the $RMSE_{R2}$ values for a range of pixel overlaps using the 2009-2018 mean $f_{std}$ value of 0.08 and the 2009-2018 maximum $f_{std}$ value of 0.2. For the maximum $f_{std}$ with pixel overlaps between 6-12 the $RMSE_{R2}$ ranges from 0.10-0.12.

## 3 Results and Discussion

### 3.1 RADARSAT-2 $f_p$ spatial and temporal variability from 2009-2018

       The spatial distribution of mosaicked RADARSAT-2 $f_p$ and pre-melt season (i.e. April) and sea ice stage of develop conditions in the CAA for the 2009-2018 time period are shown in Figures 3 and 4, respectively. Lower $f_p$ values are located primarily in the northern regions of the CAA (Queen Elizabeth Islands), Viscount-Melville Sound and the M'Clintock





Channel where the majority of the CAA's MYI is typically found. The shallow bays and narrow channels located throughout
the CAA exhibit high $f_p$ values and these regions are typically associated with smooth FYI whereas rougher ice regions (i.e.
Gulf of Boothia) are associated with lower $f_p$ values. We should expect a lower $f_p$ over MYI regions compared to FYI regions
(Grenfell and Perovich, 2004; Perovich and Polashenski, 2012) and indeed the overall spatial distribution of RADARSAT-2
$f_p$ is in excellent agreement with the spatial distribution of sea ice stage of development prior to the melt season for all years.

Figure 5a shows the time series of RADARSAT-2 $f_p$ variability together with mean April MYI area in the CAA
from 2009-2018. Over the 10-year record, the mean RADARSAT-2 $f_p$ was 0.47 and ranged from a low of 0.43 in 2009 to a
high of 0.52 in 2013. The temporal variability in RADARSAT-2 $f_p$ is reflected in the variability of April MYI area within the
CAA over the 10-year record with a statistically significant detrended correlation (R) of R=-0.89. The RADARSAT-2 $f_p$
linkage with April MYI area is particularly evident from 2011 and 2012 which were very light sea ice years within the CAA
whereby a considerable amount of the CAA's MYI area was lost during the summer melt season (Howell et al., 2013) and
this resulted in 2012 and 2013 (i.e. the years following extreme melt) being the two highest RADARSAT-2 $f_p$ years from
2009-2018 (Figure 3d-e). MYI area within in the CAA then increased following these light ice years and RADARSAT $f_p$
began to respond accordingly. In fact, there has always been a period of MYI recovery following light ice years with either
MYI grown *in situ* and/or advected from Arctic Ocean into the CAA and gradually migrating to the CAA's southern regions
(Howell et al., 2013). Figure 5b illustrates the standard deviation of RADARSAT-2 $f_p$ from 2009-2018 and spatially reflects
the process of MYI flowing southward through the CAA as RADARSAT-2 $f_p$ was more variable in the MYI regions of the
CAA compared to regions where FYI dominates the regional icescape.

What is interesting from the time series in Figure 5a is that from 2014-2018, with the exception of 2016, there was
more MYI area in April compared to 2009 yet the RADARSAT-2 $f_p$ was not as low as in 2009. In addition, 2017 and 2018
also exhibited a larger spatial coverage of MYI compared to 2009 (Figure 4a, 4i-j). We suggest that higher RADARSAT-2 $f_p$
in recent years is a result of Arctic Ocean MYI entering the CAA being younger and thinner than in 2009 (Howell and
Brady, 2019) with smoother surface topography, thereby having a higher summer melt pond coverage (Landy et al., 2015).
This seems to be particularly evident particularly in the Viscount-Melville Sound and M'Clintock Channels regions when
comparing 2009 (Figure 3a) with 2017 (Figure 3i) and 2018 (Figure 3j). Indeed, several studies have reported considerable
decreases in the age and thickness of Arctic Ocean MYI north of the CAA in recent years (e.g. Kwok, 2018; Petty et al.,
2020; Tschudi et al., 2020)

### 3.2 Comparison of RADARSAT-2 $f_p$ with and Sentinel-1, *in situ* and MODIS

Frequency distributions of RADARSAT-2 $f_p$ and Sentinel-1 $f_p$ from Scharien et al. (2017) in the CAA for 2016 and
2017 are shown in Figure 6. Sentinel-1 appears to estimate more regions of lower $f_p$ compared to RADARSAT-2 which are
typically associated with MYI. Whereas, RADARSAT-2 estimates more regions of higher $f_p$ which are typically associated
with FYI. Overall, there is good agreement between both sensors, as expected since both are C-band SAR with nearly
identical frequencies.





The *in situ* evolution of $f_p$ over FYI within the CAA acquired by Landy et al. (2014) and illustrated in Figure 7 allows us the place the RADARSAT-2 $f_p$ estimates within the melt pond stages of development classification system.

Unfortunately, no MODIS $f_p$ observations are located in close proximity to the *in situ* observations. The evolution of melt ponds on the surface of the sea ice has been classified into four distinct and consecutive stages. A brief description is provided here, and the reader is referred to Eicken et al. (2004) and Polashenski et al. (2012) for a more comprehensive description. In stage I, meltwater from snow melt fills topographic depressions on the surface of the sea ice until the ponds reach their maximum areal extent. In stage II, melt pond coverage decreases due to horizontal water transport into

macroscopic flaws and drainage through the ice. In stage III, the melt ponds typically drain through to the ocean and further changes in melt pond coverage depend on changes in surface topography and freeboard. Finally, in stage IV, melt ponds that survived the melt season refreeze and snow begins to accumulate on their surface. For 2011, RADARSAT-2 $f_p$ corresponds to the end of stage I and beginning of stage II thus providing a very good representation of the seasonal peak of the $f_p$, when the melt pond control on heat uptake and ice decay, through the ice-albedo feedback, is greatest. For 2012, RADARSAT-2 $f_p$

also corresponds to the end of stage I and beginning of stage II but is ~0.1 lower than *in situ* $f_p$ values. This is likely due to the very high maximum $f_p$ of 0.78 in 2012 as Scharien et al. (2017) found that equation (1) sometimes underestimates very high $f_p$ due to the low $\gamma^o$ signal associated with very smooth FYI.

The seasonal time series of the 8-day composite MODIS $f_p$, the maximum seasonal MODIS $f_p$ and the predicted RADARSAT-2 $f_p$ for 2009-2011 is shown in Figure 8. MODIS $f_p$ observations within the CAA indicate initial pond

formation occurred in May for all years with peak $f_p$ being reached in mid-July for 2009 and in early June for 2010 and 2011. Compared to the RADARSAT-2 $f_p$ values, the peak MODIS $f_p$ is ~0.09 smaller. We suggest this is likely because the predicted RADARSAT-2 $f_p$ corresponds to the stage of the seasonal melt pond evolution cycle when $f_p$ it is at its peak for each pixel within the CAA. The MODIS $f_p$ observations are determined weekly using 8-day composite image products that would include some melt pond formation and drainage processes prior-to, and after, the seasonal peak. Also, MODIS $f_p$

observations give the time series of $f_p$ therefore even the highest seasonal estimated MODIS $f_p$ is reduced because while some regions of the CAA are at their seasonal peak but others are behind or ahead. Therefore, we also calculated the maximum $f_p$ from MODIS regardless of timing during the melt season, for each pixel, also in Figure 7. These values more closely compare with the RADARSAT-2 $f_p$ but are still ~0.05 smaller on average.

**3.3 Influence of RADARSAT-2 $f_p$ on summer sea ice conditions**

In order to investigate if RADARSAT-2 $f_p$ values can be used to provide predictive information for summer sea ice area within the CAA, we separated the CAA into numerous predefined subregions and then determined the detrended correlations between RADARSAT-2 $f_p$ and weekly sea ice area from the CISDA regional ice charts in each region over the period of 2009-2018. We tested each week from the start of June to the end of September. The strongest correlation, together

with the corresponding week of occurrence are shown in Figures 9a and 9b, respectively. All the strongest correlations are negative, indicating – as expected – that years with higher predicted $f_p$ values are associated with lower sea ice area at a later





point in the summer. Higher $f_p$ lower the area-averaged albedo of the ice surface leading to accelerated melt and lower sea ice concentrations (e.g. Perovich and Polashenski, 2012). There is considerable spatial variability in the strongest correlation across the CAA with relatively low correlations in the majority of the northern CAA and very low correlations in the eastern

regions of the CAA. The regions of Kellet-Crozier (R=-0.92), Viscount-Melville Sound (R=-0.73), M'Clintock Channel (R=-0.77) and Norwegian Bay (R=-0.78) all exhibit statistically significant correlations above the 95% confidence level. In terms of timing for the statistically significant regions, RADARSAT-2 $f_p$ correlated the strongest to weekly sea ice area in August for all regions except Norwegian Bay (Figure 9b). Compared to previous studies, the primary difference between using $f_p$ values to predict summer sea ice conditions seems to be the timing of when the correlation is the strongest. Using

simulated $f_p$ values, Schröder et al. (2014) found the strongest correlation to September sea ice occurred for the May $f_p$. Liu et al. (2015) used observed MODIS $f_p$ values and reported the strongest correlation to September sea ice in late July. Our findings suggest that methods such as these may be able to predict August sea ice area from $f_p$ simulations or observations with higher confidence than September ice area, at least in the CAA.

Why is the relationship stronger in some regions of the CAA and weaker in others? RADARSAT-2 $f_p$ values are

determined from imagery acquired in April when ice conditions in the CAA are landfast (immobile) and do not evolve in concert with sea ice dynamics operating within the CAA. As a result, RADARSAT-2 $f_p$ values will not be spatially representative of the region's ice conditions when region-specific dynamic breakup processes dominate over thermodynamics (i.e. *in situ* melt). In other words, the origin of the some of the ice in these regions during the summer melt season will be not always be the same as in April (i.e. pre-melt) when the initial RADARSAT-2 $f_p$ value was determined.

The time series of weekly detrended RADARSAT-2 $f_p$ and weekly sea ice area for selected regions within the CAA is shown in Figure 10 and provides evidence for this regional dichotomy. In the Viscount-Melville Sound and M'Clintock regions the correlations gradually get stronger, reaching a peak in August. These regions are known to be immobile and stagnant (e.g. Melling, 2002) with the majority of breakup taking place in September which is when the relationship begins to degrade. The Kellet-Crozier is another stagnant region which supports that in the absence of considerable ice dynamics the

relationship between RADARSAT-2 $f_p$ and sea ice area is strong throughout the melt season. The time series in Penny Strait illustrates how the correlation gradually increase but when the region's dynamic break-up begins in July, ice is advected southward which degrades the correlation. This was also the case for other many regions in the northern CAA (not shown) as the flushing of sea ice southward from the northern CAA is a regular occurrence during the melt season (Melling, 2002; Howell et al., 2006). The low correlations in the south eastern regions of the CAA are also likely a function of ice dynamics

as these regions of the CAA are known to be considerably influenced by currents and wind (Prinsenberg and Hamilton, 2005) and sea ice speed in Lancaster Sound and Barrow Strait can reach 10 km day$^{-1}$ (Agnew et al., 2008).

The strong and statistically significant correlation in the Viscount-Melville Sound region is encouraging as it is a key shipping region in the northern route of the Northwest Passage. To that end, we used linear regression to predict mean August sea ice area within Viscount-Melville Sound with the detrended RADARSAT-2 $f_p$ values as a predictor. Figure 11

illustrates the results as compared to observations (detrended) from the CISDA ice charts for 2009-2018. There is reasonable





agreement between the predicted and observed sea ice area in the region with an RMSE of $18\times10^3$ km$^2$ and an R$^2$=0.44. The largest discrepancies occurred for 2013 and 2014 with the RADARSAT-2 $f_p$ model prediction resulting in too little sea ice area. Overall, within the Viscount-Melville Sound region of CAA exists period for which a significant statistical relationship exists between RADARSAT-2 $f_p$ and the summer ice area before dynamic ice motion begins to corrode the relationship.


## 4 Conclusions

In this study we predicted and analyzed spring $f_p$ using RADARSAT-2 within the CAA from 2009-2018. The spatial variability in RADARSAT-2 $f_p$ was found to be excellent agreement with the spatial distribution of sea ice stage of development prior to the melt season as high (low) $f_p$ values were associated with FYI (MYI) types. The temporal variability
of RADARSAT-2 $f_p$ over the 10-year record was significantly correlated to April MYI area, highlighting the importance of MYI within the CAA.

RADARSAT-2 $f_p$ was found to be in good agreement with the $f_p$ maximum extent observed *in situ* for 2011 but were slightly lower than 2012 when peak $f_p$ was very large (> 0.7). Compared to peak MODIS $f_p$ values, RADARSAT-2 $f_p$ values were larger by ~0.05. Based on our *in situ* comparison, RADARSAT-2 $f_p$ maybe more representative of peak $f_p$ within
the CAA compared to the MODIS 8-day product that may capture a more time-averaged $f_p$. We also found excellent agreement between RADARSAT-2 and Sentinel-1 which suggests that combining both Sentinel-1 and the recently launched RADARSAT Constellation Mission (RCM) could facilitate pan-Arctic $f_p$ estimates. The RCM will also facilitate continued investigation of additional metrics that when combined with $\gamma°$ could further improve predicted $f_p$.

The results presented in this study also indicate that RADARSAT-2 $f_p$ can provide predictive information about
summer sea ice area in certain regions of the CAA. Specifically, the strong and statistically significant de-trended correlation in the Viscount-Melville Sound region demonstrates that RADARSAT-2 $f_p$ estimates are useful for providing predictive information about summer sea ice area in the northern route of the Northwest Passage. This information could find utility in constraining regional model simulations (e.g. Lemieux et al., 2016). Alternatively, it could be advantageous to exploit the high spatial resolution of SAR and investigate if local-scale $f_p$ estimates could enhanced knowledge of summer ice conditions
in northern communities (e.g. Cooley et al., 2020). Ultimately, imagery from RCM will ensure our time series of RADARSAT-2 $f_p$ estimates in the CAA will continue, gradually building statistics facilitating the development of more robust statistical relationships in upcoming years.

## Data Availability

RADARSAT-2 imagery is available online for a fee from the Earth Observation Data Management System (https://www.eodms-sgdot.nrcan-rncan.gc.ca). RADARSAT-2 derived melt pond fraction is available through the lead author SELH (stephen.howell@canada.ca). MODIS Arctic melt pond cover fractions dataset available from the Integrated Climate Data Center (ICDC, https://icdc.cen.uni-hamburg.de/). The CISDA is available online from the Canadian Ice Service (CIS; https://www.canada.ca/en/environment-climate-change/services/ice-forecasts-observations/latest-



conditions/archive-overview.html). In situ melt pond data is available through contributing author JL (jack.landy@bristol.ac.uk)

**Author contributions**

SELH wrote the manuscript with input from all authors. SELH and MB preformed the analysis.


**Competing interests**

The authors declare that they have no conflict of interest.

**Acknowledgements**

Funding for RKS provided by Polar Knowledge Canada Science and Technology Program Grant NST-1718-0024 and Marine Environmental Observation Prediction and Response Network (MEOPAR) project 1-02-02-012.5. JL is supported by the European Space Agency Living Planet Fellowship "Arctic-SummIT" under Grant ESA/4000125582/18/I-NS and the Natural Environment Research Council Grants "PRE-MELT" NE/T000546/1 and "Diatom-ARCTIC" NE/R012849/1.

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





**Table 1.** Number of RADARSAT-2 images acquired over the Canadian Arctic Archipelago in April for 2009-2018.

| Year | RADARSAT-2 Image Count |
|------|------------------------|
| 2009 | 90 |
| 2010 | 138 |
| 2011 | 149 |
| 2012 | 149 |
| 2013 | 188 |
| 2014 | 159 |
| 2015 | 133 |
| 2016 | 159 |
| 2017 | 151 |
| 2018 | 144 |







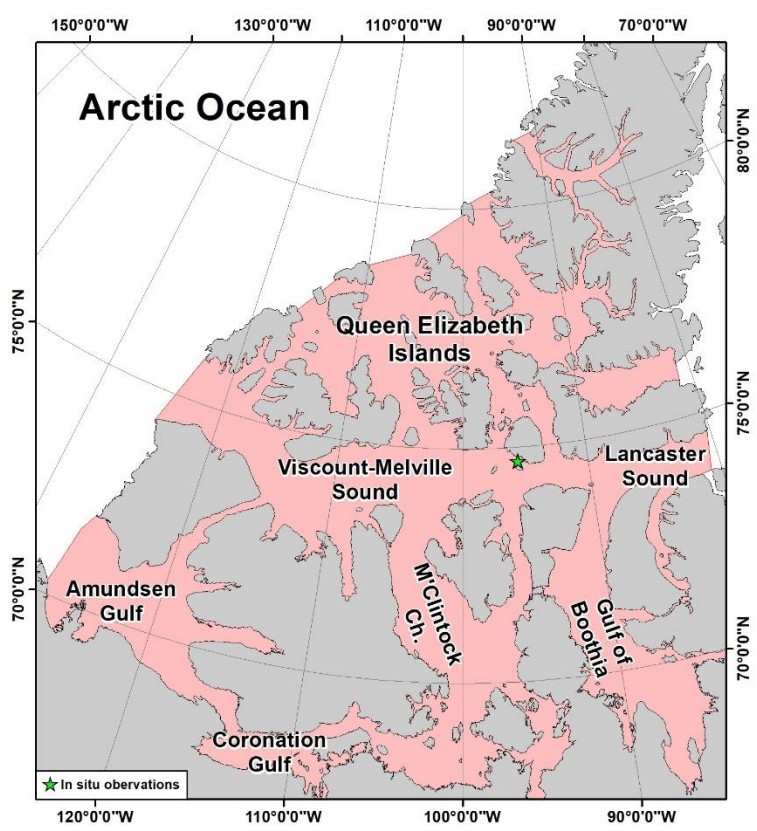

**Figure 1.** Map of the Canadian Arctic Archipelago region (red shading).


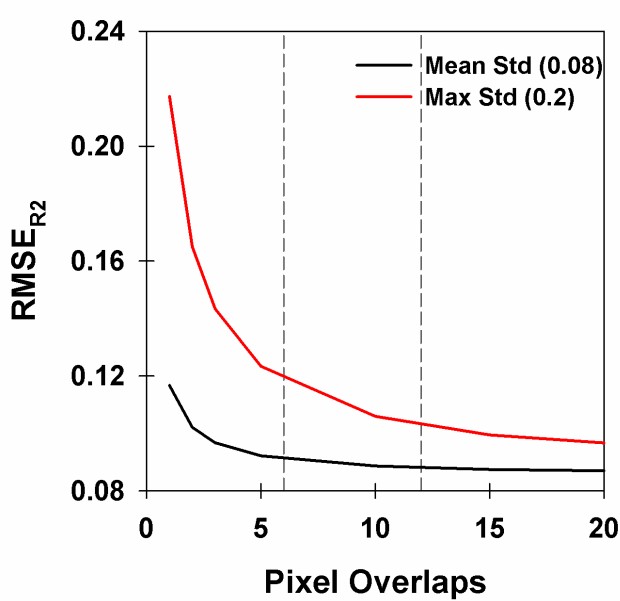

**Figure 2.** The root-mean square error of RADARSAT-2 melt pond fraction values (RMSE$_{R2}$) with increasing number of RADARSAT-2
pixel overlaps. The vertical dashed lines indicate the range of typical overlap from 2009-2018.


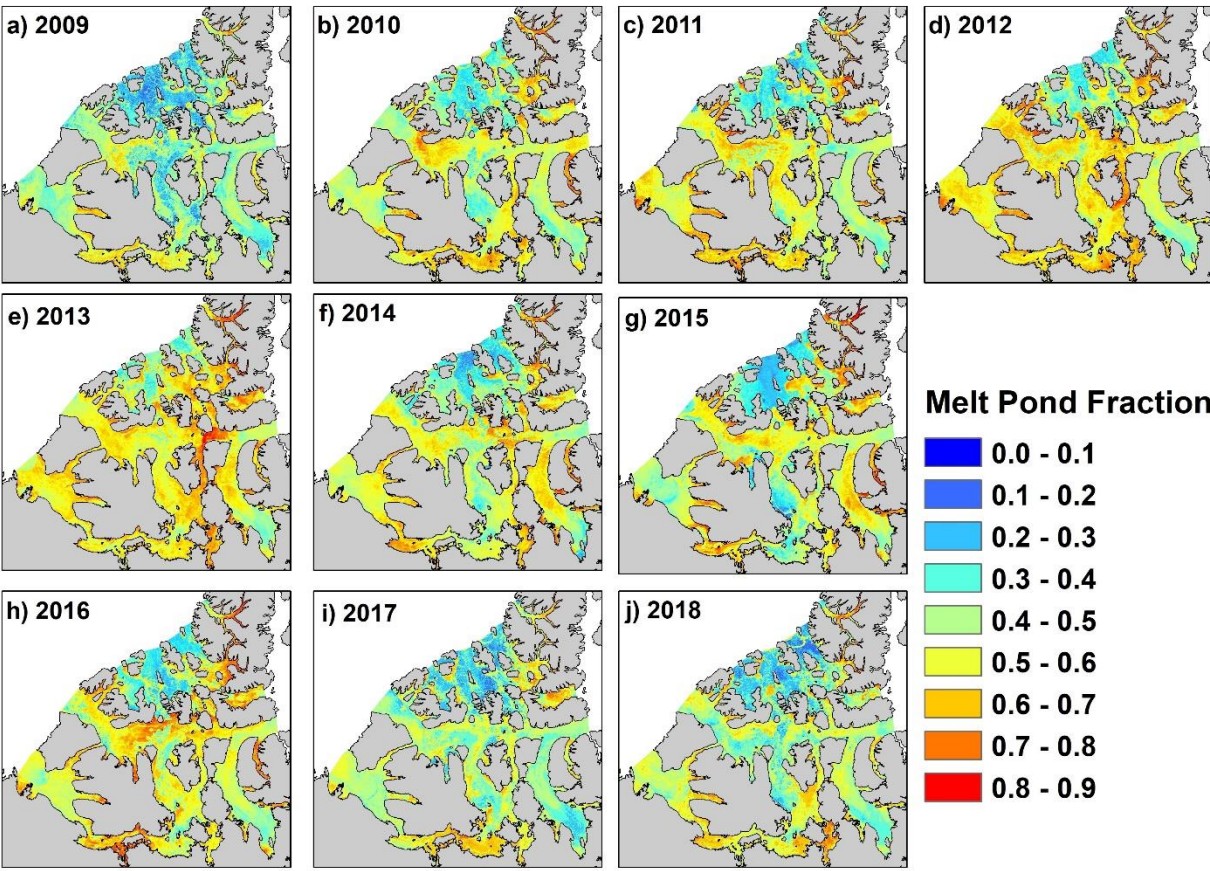

**Figure 3.** Spatial distribution of RADARSAT-2 melt pond fraction ($f_p$) in the Canadian Arctic Archipelago from 2009-2018 (a-j).






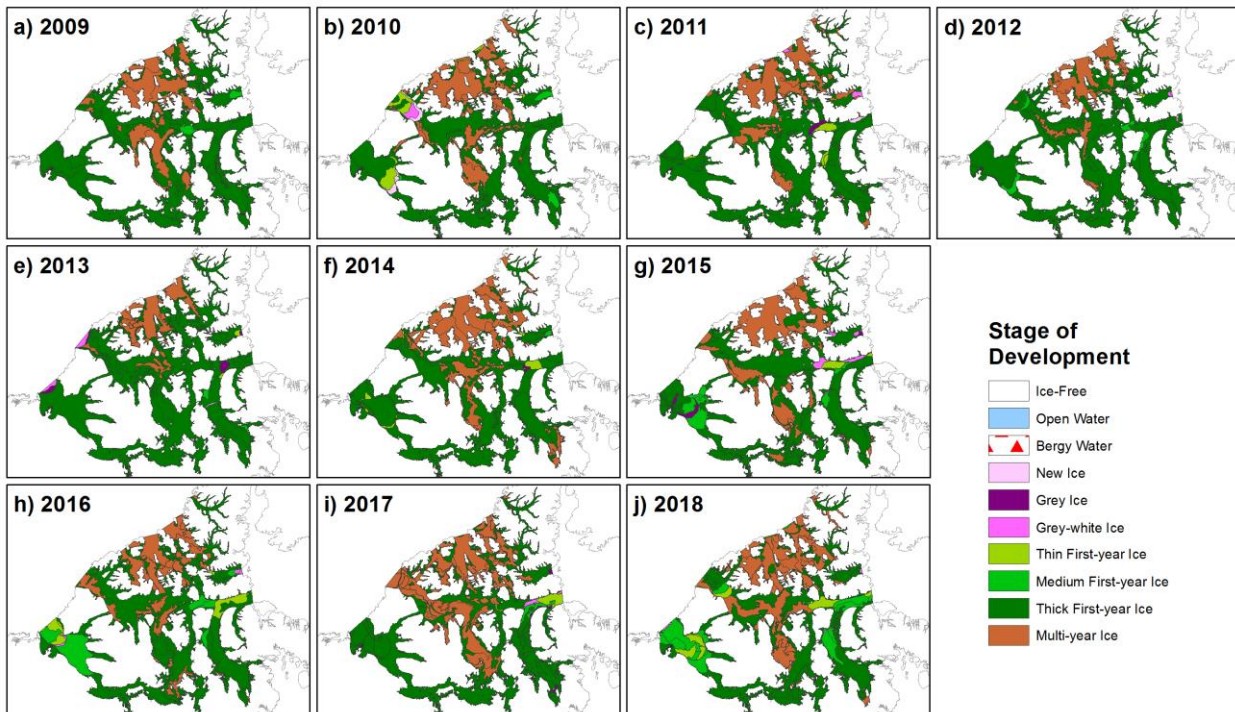

**Figure 4.** Spatial distribution of sea ice stage of development (type) on the first week of April in the Canadian Arctic Archipelago for 2009-2018 (a-j).






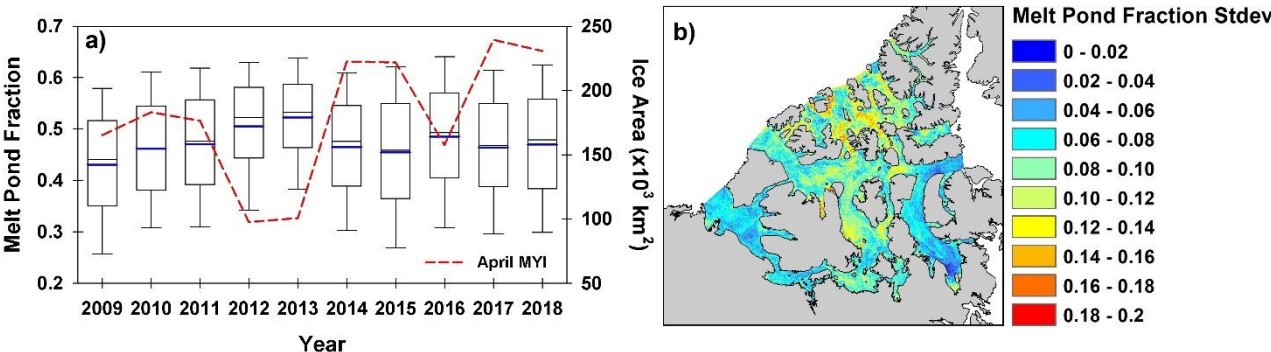

**Figure 5.** Boxplot time series of RADARSAT-2 melt pond fraction ($f_p$) and mean April multi-year ice (MYI) area in the Canadian Arctic Archipelago for 2009-2018. The solid blue line represents the mean (a). Spatial distribution of the RADARSAT-2 $f_p$ standard deviation from 2009-2018 (b).





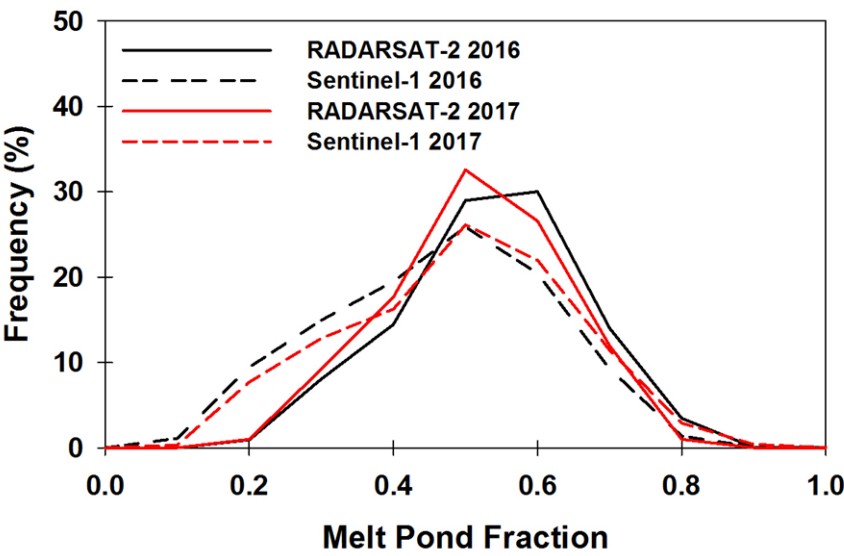


**Figure 6.** Frequency distribution (%) of RADARSAT-2 melt pond fraction ($f_p$) and Sentinel-1 $f_p$ from Scharien et al. (2017) in the Canadian Arctic Archipelago for 2016 and 2017.





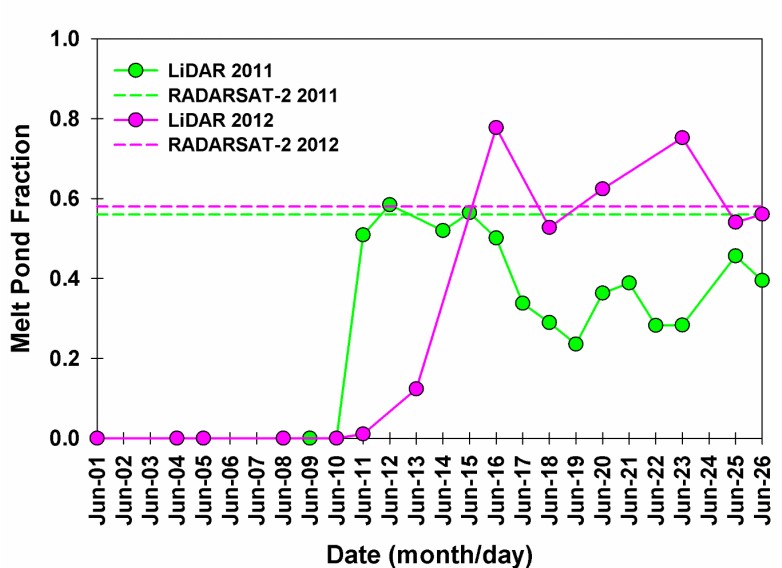

**Figure 7.** Temporal evolution of observed melt pond fraction ($f_p$) and RADARSAT-2 $f_p$ at *in situ* observations sites for 2011 (74.7229°N; -95.1763°W) and 2012 (74.7264°N; -95.5772°W).



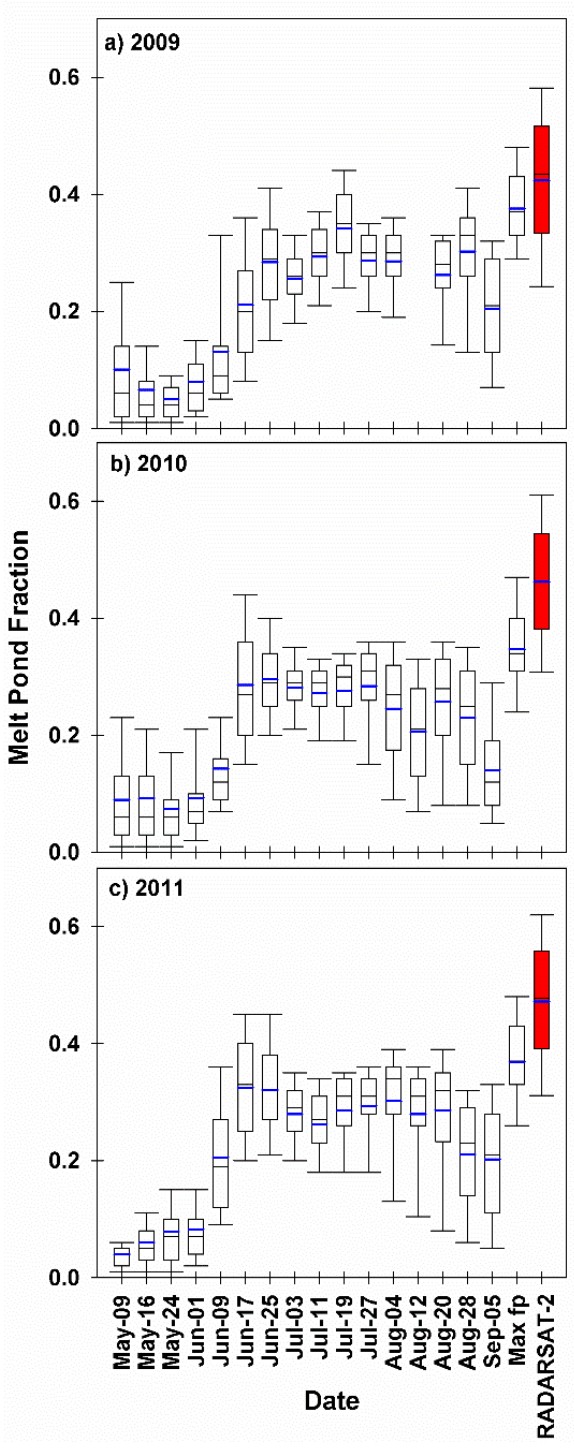


**Figure 8.** Boxplots of the seasonal time series of MODIS melt pond fraction ($f_p$), the maximum seasonal MODIS $f_p$ and RADARSAT-2 $f_p$ for (a) 2009, (b) 2010 and (c) 2011. The solid blue line represents the mean.





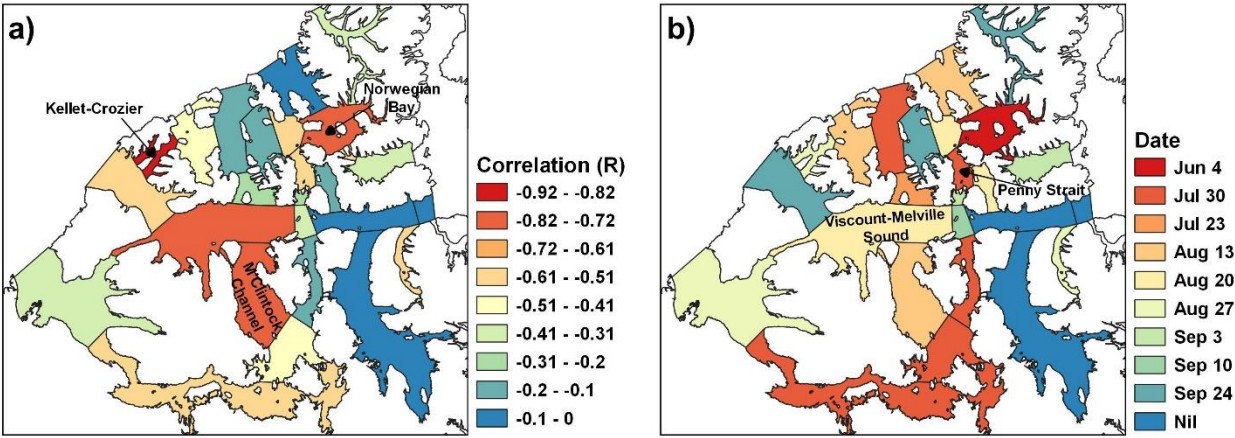

**Figure 9.** Spatial distribution of the (a) strongest detrended correlation (R) between RADARSAT-2 melt pond fraction ($f_p$) and weekly sea
ice area and (b) week of occurrence.



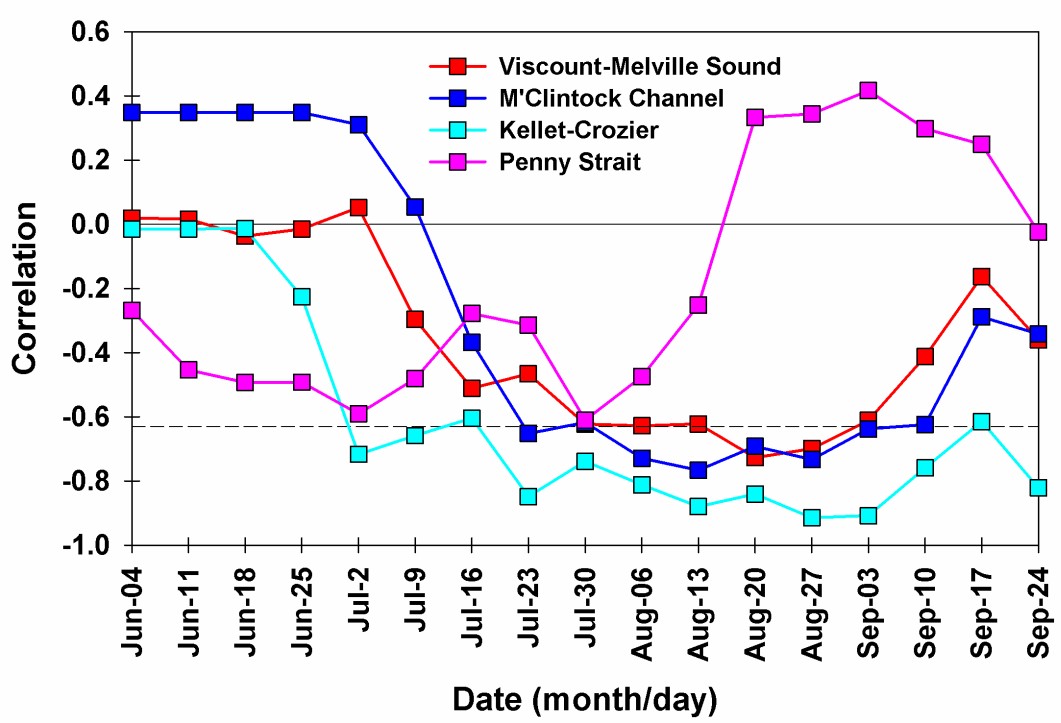

**Figure 10.** Time series of detrended correlations between RADARSAT-2 melt pond fraction ($f_p$) and weekly sea ice area for selected regions in the Canadian Arctic Archipelago from June to September. The dashed black line is statistical significance at the 95% confidence level.

580

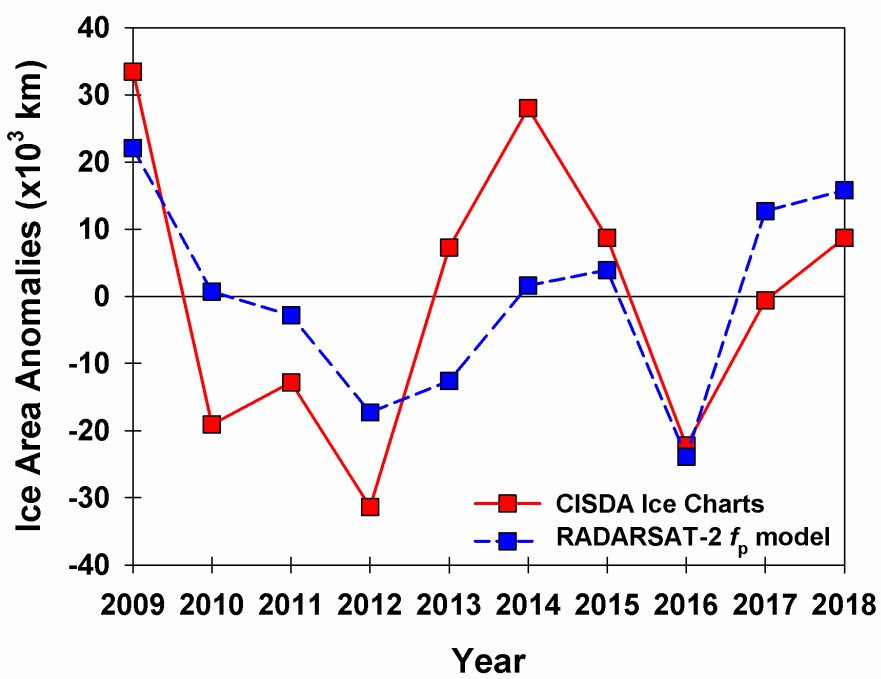

**Figure 11.** Predicated sea ice area anomalies (detrended) using RADARSAT-2 melt pond fraction ($f_p$) and observed sea ice area anomalies (detrended) from the Canadian Ice Service Digital Archive (CISDA) ice charts in the Viscount-Melville Sound region of the Canadian Arctic Archipelago, 2009-2018.