# Peer review of "Spring melt pond fraction in the Canadian Arctic Archipelago predicted from RADARSAT-2"

_The Cryosphere, 2020_

## Referee Comment (RC1) · Anonymous Referee #1 · 10 Aug 2020

This manuscript uses RADARSAT-2 imagery to derive peak melt pond fraction values for sea ice in the Canadian Arctic Archipelago between 2009 and 2018. The basic method for deriving peak pond fraction was developed in an earlier publication, and this work applies that method to a larger dataset from a different satellite. The manuscript is well written and has only a few grammatical errors that are noted below. The results presented offer valuable insight into sea ice trends and variability in the CAA. However, there are a few issues with the validation of the RADARSAT-2 derived data that should be fixed or clarified prior to publication.

General Comments

[Figure]

- You define fp as melt pond fraction. Throughout the paper you also use fp to refer to peak melt pond fraction calculated from RADARSAT-2. It would improve clarity to separate the notation for these two different parameters.

- There are two issues with the in-situ comparison:
  1. The spatial footprint of the LIDAR scans from Landy et al., (2014) are small in comparison to the 100m resolution of RADARSAT-2 data used. These in-situ datasets would only cover 1-2 pixels in the radar image. Does this area represent the whole region? Perovich (2002) determined the aggregate scale (area at which a sample can be considered representative of the larger region) at SHEBA to be multiple kilometers. If the aggregate scale is much lower in the CAA (more homogeneous ice cover) this should be discussed.

  2. Two in-situ samples are not enough to assess the accuracy of this method given the error presented in Figure 7. Here the prediction for 2011 is correct and the prediction for 2012 is not. On line 180 you state that the error is 0.1, but it looks more like 0.2 in the figure. Have you considered other in-situ datasets? For example, the three years of melt pond fraction timeseries observed on landfast ice near Utqiagvik, AK described in Polashenski et al., (2012)?

- Lines 183-194: What is the conclusion from the comparisons with MODIS? You note the reasons why RADARSAT-2 derived fp and MODIS fp could be misaligned (i.e. that the MODIS product is an 8-day average and peak ponding occurs on short timescales), and I am left with the impression that the MODIS data do not agree with your results. I would suggest expanding or clarifying the statistical analysis here. In Figure 8, both 2010 and 2011 make the RADARSAT-2 look

statistically different than MODIS. The mean (blue line) of RADARSAT-2 is approximately equal to the max (top whisker) of MODIS.

Specific comments

104 – Maybe this is covered in the Scharien paper, but is there a hypothesis for why this correlation exists? Is this method essentially just relating surface roughness (via radar backscatter) to peak pond fraction?

107 – If fp is calculated directly from each radar pixel value (Eqn. 1), how does speckle filtering impact the fp results?

165 – If both sensors are the same frequency, why is there any difference here (Figure 6) (spatial resolution difference? Sensor measurement errors?)

180 – this looks like it is 0.2 lower (difference between dashed pink line and peak pink dot). Am I reading this plot incorrectly?

248 – "Slightly lower" is maybe an understatement? It is 20% lower. Either way, quantify the amount it is lower here.

251 – In 214-231 you posit that the predictive power of this method only holds for landfast ice (i.e. when ice breakup is due to thermodynamics and not due to ice motion), how would this method be applicable to pan-Arctic estimates?

Technical Corrections

59-61 – Run-on sentence.

97 – "during April in within the CAA": Extra "in" here.

152 – This sentence is unclear.

154 – "in addition" and "also" are redundant here.

161 – 3.2 header has extra "and". Also consider including oxford comma in this list for added clarity.

183 – Again a stylistic choice, but I find oxford commas to be helpful for clarity.

190 – "but" is an extra word here.

192 – Do you mean Figure 8 here? 215 – "The origin of the some of the ice" extra

words here.
239 – "Overall, within the. . . ": Revisit sentence structure here.
253 – "Was found to be excellent agreement": Missing "in" here.
249 – "maybe" should be "may be" in this context.

References
Landy, J., Ehn, J., Shields, M. and Barber, D.: Surface and melt pond evolution on landfast first-year sea ice in the Canadian Arctic Archipelago, J. Geophys. Res. Ocean., 119(5), 3054–3075, doi:10.1002/2013JC009617, 2014.

Perovich, D. K.: Aerial observations of the evolution of ice surface conditions during summer, J. Geophys. Res., 107(C10), 8048, doi:10.1029/2000JC000449, 2002.

Polashenski, C., Perovich, D. and Courville, Z.: The mechanisms of sea ice melt pond formation and evolution, J. Geophys. Res. Ocean., 117(C1), n/a-n/a, doi:10.1029/2011JC007231, 2012.

---

## Referee Comment (RC2) · Anonymous Referee #2 · 20 Aug 2020

The manuscript uses RADARSAT-2 data to estimate melt pond fraction within the Canadian Arctic. The manuscript is clear and well written with figures clearly supporting the presented results and the discussion.

I found the investigation into the correlation between the different regions and the melt pond fraction one of the most important findings of this study. Maybe this finding could be more explicitly stated in the abstract and also in the conclusion? "Static/stable sea ice regions showed a higher detrended correlation." The mentioning of several regions is a bit vague.

Single pol RADARSAT-2 data was used, why is that? Was the combination of HH + HV

lacking? Or did the HH-channel contribute sufficient information? This may have been covered in earlier work by e.g. Scharien et al., but would then be worth reiterating.

The in-situ area only covers areas with a relatively high proportion of melt ponds, were any other in-situ data available that could be used for the validation with a smaller proportion of melt ponds? Moreover, the area covered for the in-situ data is rather small compared to the pixel size of the RADARSAT-2 images. Are there larger datasets, either more locations or covering a larger area that could be used to strengthen the argument?

The comparison between the results using Sentinel-1 and RADARSAT-2 imagery was interesting, but a discussion about why the results are different (e.g. Fig 6) is missing. Both of the images being C-band SAR one would expect the results to align quite well. Please discuss this. The comparison between the RADARSAT-2 and MODIS data, particularly figure 8, seems to suggest large differences between the two sensors, where even the maximum fp is significantly lower than the RADARSAT-2 estimates. Were there regions in the CAA that showed better agreement between the MODIS and RADARSAT-2 estimates?

Specific comments

Consider moving the information about stages of lake evolution on page 6 to the information about data or similar instead. Readers unfamiliar with melt pond development would be aided by an earlier introduction to the different stages. On P3 it is stated that the evolution stages covered by the field work covers 3 out of 4 stages, but on P 6 R177-179 it states that stage I and II was captured. Please clarify.

Is it expected that the environmental conditions remain reasonably stable in CAA during the month of April? If so maybe that could be added to strengthen the argument for combining RADARSAT-2 data for the analysis?

Minor comments

The use of the words excellent and good in the abstract are slightly abstract. Maybe it would be possible to provide some statistical measure?

P2 L41. What is the difference between sea ice area and extent? Should it possibly say sea ice type and sea ice extent?

P2 L43. Does fp here relate to maximum/mean values? Please clarify

P6. L169. Should it be . . . allows us to place the. . .?

P6. R192. Should this be Figure 8?

Fig 1. Please state what the green star indicates in the figure text.

Fig 7. Should it be -W in the coordinates.

[Figure]

---

## Author Comment (AC2) · 5 Sep 2020

**Reviewer #2**
The manuscript uses RADARSAT-2 data to estimate melt pond fraction within the Canadian Arctic. The manuscript is clear and well written with figures clearly supporting the presented results and the discussion.

**Howell et al.**
We thank this reviewer for her/his comments that have improved this manuscript. We have incorporated almost all this reviewer's suggestions.

**Reviewer #2**
I found the investigation into the correlation between the different regions and the melt pond fraction one of the most important findings of this study. Maybe this finding could be more explicitly stated in the abstract and also in the conclusion? "Static/stable sea ice regions showed a higher detrended correlation." The mentioning of several regions is a bit vague.

**Howell et al.**
Agreed.

Revised Abstract:
Dynamically stable sea ice regions within the CAA exhibited higher detrended correlations between RADARSAT-2 $f_{pk}$ summer sea ice area.

Revised Conclusions:
The results presented in this study indicate that dynamically stable sea ice regions within the CAA exhibit a higher detrended correlation between RADARSAT-2 $f_{pk}$ and summer sea ice area.

**Reviewer #2**
Single pol RADARSAT-2 data was used, why is that? Was the combination of HH + HV lacking? Or did the HH-channel contribute sufficient information? This may have been covered in earlier work by e.g. Scharien et al., but would then be worth reiterating.

**Howell et al.**
Single pol RADARAT-2 was used for two reasons. The first is that Scharien et al. (2017) found the HV data produced noisy results and the second there is not sufficient HV imagery in the early of the RADARSAT-2 to cover CAA. The latter is because only in the recent years has HH+HV been ordered operationally throughout the CAA.

Revised Data:
We limited our analysis to only RADARAT-2 at HH polarization because Scharien et al. (2017) found HV produced noisy results in addition to there not being sufficient imagery at HV polarization in the early of the RADARSAT-2 record to cover CAA.

**Reviewer #2**
The in-situ area only covers areas with a relatively high proportion of melt ponds, were any other in-situ data available that could be used for the validation with a smaller proportion of melt ponds? Moreover, the area covered for the in-situ data is rather small compared to the pixel size

of the RADARSAT-2 images. Are there larger datasets, either more locations or covering a larger area that could be used to strengthen the argument?

**Howell et al.**

Yes, we do have aerial photograph estimates of melt pond fraction obtained over and adjacent to the LiDAR site in 2012 from Scharien et al. (2014), which we have made use of to compare with RADARSAT-2 $f_{pk}$ estimates. We have added a new Figure 7b with the aerial photograph data and revised the following sections:

Revised Section 3.2

Figure 7a compares the time series of the entire 100 m LiDAR melt pond fraction coincident with the $f_{pk}$ determined from RADARSAT-2 at the coinciding pixels. For 2011, RADARSAT-2 $f_{pk}$ corresponds to the end of stage I and beginning of stage II thus providing a very good representation of the seasonal peak of the $f_p$, when the melt pond control on heat uptake and ice decay, through the ice-albedo feedback, is greatest. For 2012, RADARSAT-2 $f_{pk}$ also corresponds to the end of stage I and beginning of stage II but is ~0.1 lower than *in situ* $f_p$ values. This is likely due to the short duration but very high maximum $f_p$ of 0.78 in 2012 as Scharien et al. (2017) found that equation (1) sometimes underestimates very high $f_p$ due to the low $\gamma^o$ signal associated with very smooth FYI.

Figure 7b shows the distribution of RADARSAT-2 $f_{pk}$ and the $f_p$ determined from aerial photo observations on June 22$^{nd}$, 2012 near Resolute. The aerial photographs were acquired within 1 week of $f_{pk}$ coverage being observed at the LiDAR site. The comparison was done by averaging all RADARSAT-2 pixels within each aerial photo. The mean aerial photograph $f_p$ was 0.54 and RADARSAT-2 $f_{pk}$ was 0.53 with an the RMSE of 0.10 and bias of 0. The distributions are in reasonably good agreement but RADARSAT-2 values are slightly narrower than the distribution of $f_p$ from the aerial photographs. It is likely the RADARSAT-2 distribution is narrow on the left tail because our method captures peak pond coverage and some of the regions photographed were before or after their seasonal peak. We attribute the narrow right tail to the documented underestimation of equation (1) from Scharien et al. (2017). However, it is notable that both RADARSAT-2 and the aerial photograph datasets capture the same bimodal $f_p$ distribution, with the first mode around 0.4-0.5 characterizing rougher sea ice areas and the second mode around 0.7 capturing smooth flooded sea ice.

[Figure]

**Figure 7**. a) Temporal evolution of observed melt pond fraction ($f_p$) and RADARSAT-2 peak melt pond fraction ($f_{pk}$) at *in situ* observations sites for 2011 (74.7229°N; 95.1763°W) and 2012 (74.7264°N; 95.5772°W). b) Frequency distribution of RADARSAT-2 $f_{pk}$ and aerial photograph $f_p$ observations in Resolute Passage on June 22, 2012; the pink vertical link represents the mean LiDAR $f_p$ on June 22, 2012.

Revised Data
Aerial photographs of estimated $f_p$ directly over the LiDAR site and the adjacent sea ice area away from land and open water were also obtained on June 22, 2012.  The aerial photographs have a pixel resolution 0.22 m resolution, cover 750 m by 750 m. In total, 123 aerial photographs of $f_p$ were used and a complete description of the dataset is provided in Scharien et al. (2014).

Added Reference
Scharien, R. K., Hochheim, K., Landy, J., and Barber, D. G.: First-year sea ice melt pond fraction estimation from dual-polarisation C-band SAR – Part 2: Scaling in situ to Radarsat-2, The Cryosphere, 8, 2163–2176, https://doi.org/10.5194/tc-8-2163-2014, 2014.

**Reviewer #2**
The comparison between the results using Sentinel-1 and RADARSAT-2 imagery was interesting, but a discussion about why the results are different (e.g. Fig 6) is missing. Both of the images being C-band SAR one would expect the results to align quite well. Please discuss this. The comparison between the RADARSAT-2 and MODIS data, particularly figure 8, seems to suggest large differences between the two sensors, where even the maximum fp is significantly lower than the RADARSAT-2 estimates.

**Howell et al.**

We should have provided more discussion between and Sentinel-1 and RADARSAT-2 as also suggested by Reviewer #1.

Revised Section 3.2

Frequency distributions of RADARSAT-2 $f_{pk}$ and Sentinel-1 $f_{pk}$ from Scharien et al. (2017) in the CAA for 2016 and 2017 are shown in Figure 6. Sentinel-1 appears to estimate more regions of lower $f_{pk}$ compared to RADARSAT-2 which are typically associated with MYI. Whereas, RADARSAT-2 estimates more regions of higher $f_{peak}$ which are typically associated with smooth FYI. We consider these subtle differences to be primarily the result of taking the mean of all available April RADARSAT-2 imagery (Table 1) over all incidence angles in the CAA compared to only using images from Sentinel-1 within the CAA constrained to a certain incident angle range. As shown in Figure 2, the uncertainty in RADARSAT-2 $f_{pk}$ varies depending on the number of pixel overlaps (images). Overall, the $f_{pk}$ distributions are in good agreement between both sensors.

As for the MODIS product, it underestimates peak pond melt fraction in the CAA and is more representative of pond coverage at synoptic timescales. Even the maximum $f_p$ from MODIS is from an 8-day running mean of daily pond fraction estimates, so will underestimate the seasonal peak $f_p$ if the duration of peak ponding is <8 days. As also suggested by Reviewer #1 we firm up the wording here to point this out and have revised the text in 3.2 as follows:

Revised Section 3.2

The seasonal time series of the 8-day composite MODIS $f_p$, the maximum seasonal MODIS $f_p$ and the predicted RADARSAT-2 $f_{pk}$ for 2009-2011 is shown in Figure 8. MODIS $f_p$ observations within the CAA indicate initial pond formation occurred in May for all years with $f_{pk}$ reached in mid-July for 2009 and in early June for 2010 and 2011.  Compared to the RADARSAT-2 $f_{pk}$ values, the peak MODIS $f_p$ is ~0.20 smaller. RADARSAT-2 $f_{pk}$ is higher on average than MODIS because the MODIS 8-day product does not represent $f_{pk}$. The MODIS $f_p$ observations are determined weekly using 8-day composite image products that would include some melt pond formation and drainage processes prior-to, and after, the seasonal peak. Moreover, MODIS $f_p$ observations give the time series of $f_p$ therefore even the highest seasonal estimated MODIS $f_p$ is reduced because while some regions of the CAA are at their seasonal peak but others are behind or ahead. To that end, we also calculated the maximum $f_p$ from MODIS regardless of timing during the melt season, for each pixel, also shown in Figure 7. These values more closely compare with the RADARSAT-2 $f_{pk}$ but are still ~0.05 smaller on average. Even the maximum $f_p$ from MODIS is from an 8-day running mean of daily pond fraction estimates, so will underestimate the seasonal peak $f_p$ if the duration of peak ponding is <8 days. However, the top whisker of the box plot of the maximum $f_p$ from MODIS indicates that MODIS does capture some regions at peak during the 8-day time series. Although we are using MODIS $f_p$ product to compare against out RADARSAT-2 $f_{pk}$ estimates, Rösel et al. (2012) found that the MODIS $f_p$ product also has errors up to ~0.1. Overall, MODIS $f_p$ estimates are more representative of the seasonal mean $f_p$ rather than $f_{pk}$ within the CAA.

**Reviewer #2**

Were there regions in the CAA that showed better agreement between the MODIS and RADARSAT-2 estimates?

**Howell et al.**

Not really. We produced spatial maps but decided not to include them because the do not provide as much information as the boxplots.

Specific comments
**Reviewer #2**

Consider moving the information about stages of lake evolution on page 6 to the information about data or similar instead. Readers unfamiliar with melt pond development would be aided by an earlier introduction to the different stages. On P3 it is stated that the evolution stages covered by the field work covers 3 out of 4 stages, but on P 6 R177-179 it states that stage I and II was captured. Please clarify.

**Howell et al.**

We assume the reviewer means pond evolution. This seems the ideal place to describe these stages in accordance with the Figure 7 showing that the LiDAR site captures stages 1 to 3. Since the site is over first-year ice stage 4 will not occur and requires no discussion. We have removed references to the melt pond stages in the data description. Re-reading our text, it seems clear that the RADARSAT-2 $f_{pk}$ values fall within end of stage I and beginning of stage II at the LiDAR site.

**Reviewer #2**

Is it expected that the environmental conditions remain reasonably stable in CAA during the month of April? If so maybe that could be added to strengthen the argument for combining RADARSAT-2 data for the analysis?

**Howell et al.**

Yes, it is expected. We have already explicitly stated this in the methodology: "...together with the fact that the majority of the sea ice in the CAA is landfast (immobile) during April which results in a temporally stable $f_{pk}$ for all April images."

**Reviewer #2**

Minor comments
The use of the words excellent and good in the abstract are slightly abstract. Maybe it would be possible to provide some statistical measure?

**Howell et al.**

We added a statistical measure the temporal linkage but the spatial needs to be visual.

Revised Abstract
The temporal variability of RADARSAT-2 $f_{pk}$ over the 10-year record was found to be strongly linked to the variability of mean April multi-year ice area with a statistically significant detrended correlation (R) of R=-0.89. The spatial distribution of RADARSAT-2 $f_{pk}$ was found to be in excellent agreement with the sea ice stage of development prior to the melt season.

**Reviewer #2**
P2 L41. What is the difference between sea ice area and extent? Should it possibly say sea ice type and sea ice extent?

**Howell et al.**
No area and extent are the correct terms and the ones most commonly used. Sea ice area is ice concentration multiped by the area of the region. Extent is also calculated as area multiplied by ice concentration but it this assumes that the area is 100% provided it is greater than a certain threshold (i.e. typically 15%). A great explanation is found on the NSIDC website "A simplified way to think of extent versus area is to imagine a slice of swiss cheese. Extent would be a measure of the edges of the slice of cheese and all of the space inside it. Area would be the measure of where there is cheese only, not including the holes. That is why if you compare extent and area in the same time period, extent is always bigger. A more precise explanation of extent versus area gets more complicated."
http://nsidc.org/arcticseaicenews/faq/#:~:text=The%20most%20common%20threshold%20(and, said%20to%20be%20ice%20free.

**Reviewer #2**
P2 L43. Does fp here relate to maximum/mean values? Please clarify

**Howell et al.**
As suggested by Reviewer #1 we have modified the notation throughout the manuscript to denote melt pond fraction as $f_p$ and peak melt pond fraction as $f_{pk}$.

**Reviewer #2**
P6. L169. Should it be : : : allows us to place the: : :?

**Howell et al.**
Yes. Inserted "to".

**Reviewer #2**
P6. R192. Should this be Figure 8?

**Howell et al.**
Yes.

**Reviewer #2**
Fig 1. Please state what the green star indicates in the figure text.

**Howell et al.**
New Figure caption as follows:
Figure 1. Map of the Canadian Arctic Archipelago region (red shading). The green star indicates the location of the LiDAR and aerial photograph observations.

**Reviewer #2**
Fig 7. Should it be -W in the coordinates.

**Howell et al.**
Removed the '-'

---

## Referee Comment (RC3) · Anonymous Referee #3 · 6 Oct 2020

This paper derives the melt pond faction in month of April derived from Radarsat-2 imagery to predict the resulting sea ice area over the ensuing summer melt season within the Canadian archipelago, from years 2009-2018. The best results were found to be between stage of development in April and melt pond fraction, following the related paper by Scharien et al., 2017. Other comparisons were more challenging but were well explained.

Due to my tardiness with this review, which I apologize for, I did read the other two reviews and the authors' response to both. I generally agreed with the reviewers comments and the responses were well posed. I will only add a couple of additional com-

ments, that may be a little different.

1. Figure 7. As with the other two reviewers, I had some concerns with this figure, due to the relatively limited area of the lidar observations. The inclusion of the aerial photography and SAR comparisons that were added in Figure 7b are a valuable addition. Going back to Figure 7a, the Radarsat results themselves have no response to the changing melt conditions before, during and after. There is little change between the two years. Before the addition of Fig. 7b, I was thinking of not including it. I now wonder if they included a few more surrounding pixels to examine, like a 3X3 window, some variation might appear. How many R2 frames were examined during the field measurements periods?

2. Section 3.2, first paragraph regarding R2 and Sentinel-1. Please add that S1 data collections for sea ice nominally also use HH polarization, same as R2. I am wondering about differences in the noise floor and SNR between the two systems that may be leading to some of the differences seen in Fig. 6. Were an approximately equivalent number of images used by both sensors about the same or different, thinking about Fig. 2?

3. Modis comparisons with R2, section 3.2 and Fig. 8. Please specify the resolution for the Modis products. What is the sensitivity of Modis to melt pond size? If one makes the assumption that Modis may not detect smaller ponds, that by itself may account for the differences seen in Modis Max pond fraction and R2 results, couldn't it? Also the 8-day composite of Modis may limit small pond fraction. Please clarify the impact of Modis resolution on pond fraction.

4. Regarding Figs. 3 and 4 and Fig. 9 and 10. The relationship between stage of development and pond fraction was quite clear, shown in Fig.3-4. The greatest extent of low fractions were nearly all up in the northern CAA, with more variability, higher fractions in other areas. Then you come to Fig. 9 where any possible trend that one might expect in the MY/low fraction area in the north and in other regions goes away.

The authors explain the variations in A and B, in melt pond fraction/week of strongest correlation, by dynamics, southward transport of lower pond fraction ice. The patterns in Fig3-4 were so clear and then it becomes unclear, although there is some similarity in patterns between Viscount-Melville and McClintock in Fig. 10. It's all pretty interesting and rather surprising. I urge the authors to continue to investigate this topic. Perhaps the addition of ice motion drift can provide more insight.
* * *

---

## Author Comment (AC3) · 7 Oct 2020

**Reviewer #3**

This paper derives the melt pond faction in month of April derived from Radarsat-2 imagery to predict the resulting sea ice area over the ensuing summer melt season within the Canadian archipelago, from years 2009-2018. The best results were found to be between stage of development in April and melt pond fraction, following the related paper by Scharien et al., 2017. Other comparisons were more challenging but were well explained. Due to my tardiness with this review, which I apologize for, I did read the other two reviews and the authors' response to both. I generally agreed with the reviewers comments and the responses were well posed. I will only add a couple of additional comments, that may be a little different.

**Howell et al.**
We have addressed all the Reviewer's comments. The MODIS spatial resolution suggestion was particularly useful for improving the manuscript.

**Reviewer #3**
1. Figure 7. As with the other two reviewers, I had some concerns with this figure, due to the relatively limited area of the lidar observations. The inclusion of the aerial photography and SAR comparisons that were added in Figure 7b are a valuable addition. Going back to Figure 7a, the Radarsat results themselves have no response to the changing melt conditions before, during and after. There is little change between the two years. Before the addition of Fig. 7b, I was thinking of not including it. I now wonder if they included a few more surrounding pixels to examine, like a 3X3 window, some variation might appear. How many R2 frames were examined during the field measurements periods?

**Howell et al.**
There is little variability using a 3x3 window near the LiDAR site and in this case we feel a direct one-to-one comparison is best. The individual RADARSAT-2 frames are averaged into a mosaic for the year and on average there are between 6 and 11 overlaps (Figure 2) with 8 in 2011 and 5 in 2012 over the LiDAR site. However, the point raised by the Reviewer is that it is important to mention uncertainty in the text and fewer pixel overlaps could also result in a reduction of the RADARSAT-2 peak pond fraction estimate. In 2012, the RADARSAT-2 peak melt pond fraction at the LiDAR pixel could be 0.1 higher according to Figure 2 which would be closer to the LiDAR values.

Revised Section 3.2 as follows:
This is likely due to the short duration but very high maximum $f_p$ of 0.78 in 2012 as Scharien et al. (2017) found that equation (1) sometimes underestimates very high $f_p$ due to the low $\gamma^o$ signal associated with very smooth FYI. Another consideration is the uncertainty in RADARSAT-2 $f_{pk}$ estimates is least 0.1 (Figure 2) which would bring the RADARSAT-2 $f_{pk}$ values closer to the *in situ* values.

**Reviewer #3**
2. Section 3.2, first paragraph regarding R2 and Sentinel-1. Please add that S1 data collections for sea ice nominally also use HH polarization, same as R2. I am wondering about differences in the noise floor and SNR between the two systems that may be leading to some of the differences

seen in Fig. 6. Were an approximately equivalent number of images used by both sensors about the same or different, thinking about Fig. 2?

**Howell et al.**
We do not think it is a noise floor issue but rather it is an incident angle issue as we mention explicitly in the text. There were more Sentinel-1 images used to cover the CAA than RADARSAT-2 images but they were constrained to a certain incidence angle range. This was not possible with RADARSAT-2 and to create a close-to-seamless mosaic across the CAA with RADARSAT-2 we needed to take the average of the overlapping peak melt pond fraction values. Overall, the distributions are in very good agreement despite the different approaches.

**Reviewer #3**
3. Modis comparisons with R2, section 3.2 and Fig. 8. Please specify the resolution for the Modis products. What is the sensitivity of Modis to melt pond size? If one makes the assumption that Modis may not detect smaller ponds, that by itself may account for the differences seen in Modis Max pond fraction and R2 results, couldn't it? Also the 8-day composite of Modis may limit small pond fraction. Please clarify the impact of Modis resolution on pond fraction.

**Howell et al.**
This is a very good suggestion. By itself, the MODIS product spatial resolution is unlikely to be the primary cause since the temporal domain spans 8-days but the fact that the 12.5 km grid cell is made up of smaller 500 m pixels likely at different stages of pond evolution is another reason why the peak fraction is difficult to capture with MODS. We have inserted another sentence into our revised the MODIS comparison.

Revised Section in 3.2:
Moreover, MODIS $f_p$ values are essentially aggregated from 500 m clear-sky pixels within a 12.5 km x 12.5 km grid cell (Rösel et al., 2012) and the 500 m spatial resolution may limit detection of smaller pond fractions as well as not all of the 500 m pixels within the 12.5 km x 12.5 km grid cell are likely to be at the same melt pond stage evolution.

Revised Data and Methods:
Finally, we made use of 8-day composite satellite observations of $f_p$ obtained from the MODIS Arctic melt pond cover fractions dataset that has a spatial resolution of 12.5 km for the period of 2009-2011 (Rösel et al., 2012).

**Reviewer #3**
4. Regarding Figs. 3 and 4 and Fig. 9 and 10. The relationship between stage of development and pond fraction was quite clear, shown in Fig.3-4. The greatest extent of low fractions were nearly all up in the northern CAA, with more variability, higher fractions in other areas. Then you come to Fig. 9 where any possible trend that one might expect in the MY/low fraction area in the north and in other regions goes away. The authors explain the variations in A and B, in melt pond fraction/week of strongest correlation, by dynamics, southward transport of lower pond fraction ice. The patterns in Fig3-4 were so clear and then it becomes unclear, although there is some similarity in patterns between Viscount-Melville and McClintock in Fig. 10. It's all pretty interesting and rather surprising. I urge the authors to continue to investigate this topic. Perhaps

the addition of ice motion drift can provide more insight.

**Howell et al.**
We agree and tracking the floes will likely improve the relationship which is something we are working on.  Indeed, Viscount-Melville and the M'Clintock Channel have similar patterns because they have similar ice regimes (stagnant) so it is good to see agreement between them.

---

## Author Comment (AC1)

**Reviewer #1**

This manuscript uses RADARSAT-2 imagery to derive peak melt pond fraction values for sea ice in the Canadian Arctic Archipelago between 2009 and 2018. The basic method for deriving peak pond fraction was developed in an earlier publication, and this work applies that method to a larger dataset from a different satellite. The manuscript is well written and has only a few grammatical errors that are noted below. The results presented offer valuable insight into sea ice trends and variability in the CAA. However, there are a few issues with the validation of the RADARSAT-2 derived data that should be fixed or clarified prior to publication.

**Howell et al.**
We thank this reviewer for her/his comments that have improved this manuscript considerably. We have incorporated almost all of her/his suggestions.

**Reviewer #1**
General Comments
You define $f_p$ as melt pond fraction. Throughout the paper you also use $f_p$ to refer to peak melt pond fraction calculated from RADARSAT-2. It would improve clarity to separate the notation for these two different parameters.

**Howell et al.**
Very good suggestion. We have chosen to define peak melt pond fraction as $f_{pk}$ and have changed the text throughout the manuscript to reflect this new notation.

**Reviewer #1**
There are two issues with the in-situ comparison:
1. The spatial footprint of the LIDAR scans from Landy et al., (2014) are small in comparison to the 100m resolution of RADARSAT-2 data used. These in-situ datasets would only cover 1-2 pixels in the radar image. Does this area represent the whole region? Perovich (2002) determined the aggregate scale (area at which a sample can be considered representative of the larger region) at SHEBA to be multiple kilometers. If the aggregate scale is much lower in the CAA (more homogeneous ice cover) this should be discussed.

**Howell et al.**
It is true the LiDAR areas would cover only ~1-2 pixels, however we only compared the LiDAR pond fraction to the ~1-2 RADARSAT-2 pixels directly coincident with the site. Therefore, we are not validating RADARSAT-2 melt pond fraction against a spot LiDAR *in situ* measurements, we are just validating the entire 100 m LiDAR melt pond fraction directly at the sampling site. In this case, it does not matter whether the *in situ* samples are representative of the aggregate scale. We have clarified this in text so other readers to confuse other readers:

Revised Section 3.2
Figure 7a compares the time series of the entire 100 m LiDAR $f_k$ coincident with the $f_{pk}$ determined from RADARSAT-2 at the coinciding pixels.

**Reviewer #1**

2. Two in-situ samples are not enough to assess the accuracy of this method given the error presented in Figure 7. Here the prediction for 2011 is correct and the prediction for 2012 is not. On line 180 you state that the error is 0.1, but it looks more like 0.2 in the figure. Have you considered other in-situ datasets? For example, the three years of melt pond fraction timeseries observed on landfast ice near Utqiagvik, AK described in Polashenski et al., (2012)?

**Howell et al.**

We are limited by the scarcity of *in situ* melt pond fraction observations in the CAA and would have used more if we could. Moreover, finding observations that coincide with peak pond fraction further adds to the scarcity problem and the MODIS analysis was attempt to alleviate this problem. Unfortunately, we cannot use the *in situ* melt pond fraction dataset from Polashenski et al. (2012) because our RADARSAT-2 data only has consistent coverage in the Canadian Arctic waters in accordance with the operational domain of the Canadian Ice Service and therefore the Chukchi Sea is not covered. Despite having only two *in situ* samples, they least cover a long temporal time period allowing us to test whether RADARSAT-2 picks out the seasonal mean pond fraction or peak pond fraction. However, we do have aerial photograph estimates of melt pond fraction obtained over and adjacent the LiDAR site in 2012 from Scharien et al. (2014), which we have made use of to compare with RADARSAT-2 $f_{pk}$ estimates. We have added a new Figure 7b with the aerial photograph data and revised the following sections:

Revised Section 3.2

Figure 7a compares the time series of the entire 100 m LiDAR melt pond fraction coincident with the $f_{pk}$ determined from RADARSAT-2 at the coinciding pixels. For 2011, RADARSAT-2 $f_{pk}$ corresponds to the end of stage I and beginning of stage II thus providing a very good representation of the seasonal peak of the $f_p$, when the melt pond control on heat uptake and ice decay, through the ice-albedo feedback, is greatest. For 2012, RADARSAT-2 $f_{pk}$ also corresponds to the end of stage I and beginning of stage II but is ~0.1 lower than *in situ* $f_p$ values. This is likely due to the short duration but very high maximum $f_p$ of 0.78 in 2012 as Scharien et al. (2017) found that equation (1) sometimes underestimates very high $f_p$ due to the low $\gamma^o$ signal associated with very smooth FYI.

Figure 7b shows the distribution of RADARSAT-2 $f_{pk}$ and the $f_p$ determined from aerial photo observations on June 22[nd], 2012 near Resolute. The aerial photographs were acquired within 1 week of $f_{pk}$ coverage being observed at the LiDAR site. The comparison was done by averaging all RADARSAT-2 pixels within each aerial photo. The mean aerial photograph $f_p$ was 0.54 and RADARSAT-2 $f_{pk}$ was 0.53 with an the RMSE of 0.10 and bias of 0. The distributions are in reasonably good agreement but RADARSAT-2 values are slightly narrower than the distribution of $f_p$ from the aerial photographs. It is likely the RADARSAT-2 distribution is narrow on the left tail because our method captures peak pond coverage and some of the regions photographed were before or after their seasonal peak. We attribute the narrow right tail to the documented underestimation of equation (1) from Scharien et al. (2017). However, it is notable that both RADARSAT-2 and the aerial photograph datasets capture the same bimodal $f_p$ distribution, with the first mode around 0.4-0.5 characterizing rougher sea ice areas and the second mode around 0.7 capturing smooth flooded sea ice.

[Figure]

**Figure 7**. a)  Temporal evolution of observed melt pond fraction ($f_p$) and RADARSAT-2 peak melt pond fraction ($f_{pk}$) at *in situ* observations sites for 2011 (74.7229°N; 95.1763°W) and 2012 (74.7264°N; 95.5772°W). b) Frequency distribution of RADARSAT-2 $f_{pk}$ and aerial photograph $f_p$ observations in Resolute Passage on June 22, 2012; the pink vertical link represents the mean LiDAR $f_p$ on June 22, 2012.

Revised Section 2.1
Aerial photographs of estimated $f_p$ directly over the LiDAR site and the adjacent sea ice area away from land and open water were also obtained on June 22, 2012.  The aerial photographs have a pixel resolution 0.22 m resolution, cover 750 m by 750 m. In total, 123 aerial photographs of $f_p$ were used and a complete description of the dataset is provided in Scharien et al. (2014).

Added Reference
Scharien, R. K., Hochheim, K., Landy, J., and Barber, D. G.: First-year sea ice melt pond fraction estimation from dual-polarisation C-band SAR – Part 2: Scaling in situ to Radarsat-2, The Cryosphere, 8, 2163–2176, https://doi.org/10.5194/tc-8-2163-2014, 2014.

**Reviewer #1**
Lines 183-194: What is the conclusion from the comparisons with MODIS? You note the reasons why RADARSAT-2 derived $f_p$ and MODIS $f_p$ could be misaligned (i.e. that the MODIS product is an 8-day average and peak ponding occurs on short timescales), and I am left with the impression that the MODIS data do not agree with your results. I would suggest expanding or clarifying the statistical analysis here. In Figure 8, both 2010 and 2011 make the RADARSAT-2 look statistically different than MODIS. The mean (blue line) of RADARSAT-2 is approximately equal to the max (top whisker) of MODIS.

**Howell et al.**
This is a good point raised by the Reviewer and we were not definitive in our wording based on the boxplots. The conclusion is that RADARSAT-2 pond fraction is higher on average than MODIS because the MODIS 8-day product is not representative of $f_{pk}$ in the CAA. The weekly boxplots and max MODIS pond fraction boxplot all support this conclusion. We note that the box plot of maximum $f_p$ from MODIS does capture some regions at peak during the 8-day time series. Another point is that MODIS estimation error needs to be acknowledged because although it is treated here as validation for the RADARSAT-2 $f_{pk}$ estimates and rightly so but it has its own error component. We clarify this section in here as follows:

Revised Section 3.2
  The seasonal time series of the 8-day composite MODIS $f_p$, the maximum seasonal MODIS $f_p$ and the predicted RADARSAT-2 $f_{pk}$ for 2009-2011 is shown in Figure 8. MODIS $f_p$ observations within the CAA indicate initial pond formation occurred in May for all years with $f_{pk}$ reached in mid-July for 2009 and in early June for 2010 and 2011. Compared to the RADARSAT-2 $f_{pk}$ values, the peak MODIS $f_p$ is ~0.20 smaller. RADARSAT-2 $f_{pk}$ is higher on average than MODIS because the MODIS 8-day product does not represent $f_{pk}$. The MODIS $f_p$ observations are determined weekly using 8-day composite image products that would include some melt pond formation and drainage processes prior-to, and after, the seasonal peak. Moreover, MODIS $f_p$ observations give the time series of $f_p$ therefore even the highest seasonal estimated MODIS $f_p$ is reduced because while some regions of the CAA are at their seasonal peak but others are behind or ahead. To that end, we also calculated the maximum $f_p$ from MODIS regardless of timing during the melt season, for each pixel, also shown in Figure 7. These values more closely compare with the RADARSAT-2 $f_{pk}$ but are still ~0.05 smaller on average. Even the maximum $f_p$ from MODIS is from an 8-day running mean of daily pond fraction estimates, so will underestimate the $f_{pk}$ if the duration of peak ponding is <8 days. However, the top whisker of the box plot of the maximum $f_p$ from MODIS indicates that MODIS does capture some regions at peak during the 8-day time series. Although we are using MODIS $f_p$ product to compare against our RADARSAT-2 $f_{pk}$ estimates, Rösel et al. (2012) found that the MODIS $f_p$ product also has errors up to ~0.1. Overall, MODIS $f_p$ estimates are more representative of the seasonal mean $f_p$ rather than $f_{pk}$ within the CAA.

Revised Conclusion
Based on our comparative analysis, RADARSAT-2 $f_{pk}$ is more representative of peak $f_p$ within the CAA compared to the MODIS 8-day product which on average was found to underestimate $f_{pk}$ by ~0.2 and the is more representative of the seasonal mean $f_p$.

**Reviewer #1**
Specific comments
104 – Maybe this is covered in the Scharien paper, but is there a hypothesis for why this correlation exists? Is this method essentially just relating surface roughness (via radar backscatter) to peak pond fraction?

**Howell et al.**
Yes, it is explicitly covered and exploits the basic hypothesis that winter backscatter increases with increasing topography, for FYI, and increasing volume scattering, which is related to topography, for MYI. In each case, the increased topography leads to lower pond fraction, and

visa versa. The high resolution optical imagery helps exploit this relationship. That is, using high spatial resolution optical imagery Scharien et al. (2017) were able to isolate internally coherent, and externally discrete, zones of sea ice in order to compare backscatter/texture and $f_p$ and thus create simple models.

**Reviewer #1**
107 – If fp is calculated directly from each radar pixel value (Eqn. 1), how does speckle filtering impact the fp results?

**Howell et al.**
The impact of speckle filtering/not filtering was not assessed. As with most SAR images speckle is a problem with the goal being to obtain the most representative backscatter value for a local region (i.e. a cleaner image). The Lee facilitates this by smoothing the image without removing edges or sharp features in the images while minimizing the loss of radiometric and textural information. Although speckle filtering will change the $f_{pk}$ results for specific pixels, it will not impact $f_{pk}$ at the scale of the filter (i.e. within an x by x pixel area).

**Reviewer #1**
165 – If both sensors are the same frequency, why is there any difference here (Figure 6) (spatial resolution difference? Sensor measurement errors?)

**Howell et al.**
Good point. We should have provided some explanation for these differences

Revised Section 3.2
     Frequency distributions of RADARSAT-2 $f_{peak}$ and Sentinel-1 $f_{peak}$ from Scharien et al. (2017) in the CAA for 2016 and 2017 are shown in Figure 6. Sentinel-1 appears to estimate more regions of lower $f_{peak}$ compared to RADARSAT-2 which are typically associated with MYI. Whereas, RADARSAT-2 estimates more regions of higher $f_{peak}$ which are typically associated with smooth FYI. We consider these subtle differences to be primarily the result of taking the mean of all available April RADARSAT-2 imagery (Table 1) over all incidence angles in the CAA compared to only using images from Sentinel-1 within the CAA constrained to a certain incident angle range. As shown in Figure 2, the uncertainty in RADARSAT-2 $f_{pk}$ varies depending on the number of pixel overlaps (images). Overall, the $f_{pk}$ distributions are in good agreement between both sensors.

**Reviewer #1**
180 – this looks like it is 0.2 lower (difference between dashed pink line and peak pink dot). Am I reading this plot incorrectly?

**Howell et al.**
It should be 0.19 not 0.9. We have revised it ~0.2.

**Reviewer #1**
248 – "Slightly lower" is maybe an understatement? It is 20% lower. Either way, quantify the amount it is lower here.

**Howell et al.**
Revised Section 3.2
RADARSAT-2 $f_{pk}$ was found to be in good agreement with the $f_p$ maximum extent observed *in situ* for 2011 but was ~0.2 lower than 2012 when $f_{pk}$ was very large ($> 0.7$) for a very short duration (1-2 days).

**Reviewer #1**
251 – In 214-231 you posit that the predictive power of this method only holds for landfast ice (i.e. when ice breakup is due to thermodynamics and not due to ice motion), how would this method be applicable to pan-Arctic estimates?

**Howell et al.**
In that case a Lagrangian tracking approach would be needed or the integrated melt pond fraction could be used with evolving sea ice extent. In both cases, significant testing would be required. We are working on this, but it is considerably outside the scope of this analysis.

**Reviewer #1**
Technical Corrections
59-61 – Run-on sentence.

**Howell et al.**
Revised Introduction
Model simulations have been utilized to understand the current and predicted future variability of sea ice conditions in the CAA (e.g. Dumas et al., 2006; Sou and Flato, 2009, Howell et al., 2016; Laliberté et al., 2016; Hu et al., 2018; Laliberté et al., 2018). However, modeling the CAA still remains challenging because complex sea ice dynamic and thermodynamic processes are often not accurately resolved in its narrow channels and inlets

**Reviewer #1**
97 – "during April in within the CAA": Extra "in" here.

**Howell et al.**
Removed

**Reviewer #1**
152 – This sentence is unclear.

**Howell et al.**
Revised:
What is interesting in Figure 5a is that the mean RADARSAT-2 $f_{peak}$ in 2009 was lower than all years from 2014-2018 (with the exception of 2016) despite the CAA containing less MYI area.

**Reviewer #1**
154 – "in addition" and "also" are redundant here.

**Howell et al.**
Removed "also"

**Reviewer #1**
161 – 3.2 header has extra "and". Also consider including oxford comma in this list for added clarity.

**Howell et al.**
Revised:
3.2 Comparison of RADARSAT-2 $f_{pk}$ with Sentinel-1 $f_{pk}$, in situ $f_p$, and MODIS $f_p$

**Reviewer #1**
183 – Again a stylistic choice, but I find oxford commas to be helpful for clarity.

**Howell et al.**
Revised:
The seasonal time series of the 8-day composite MODIS $f_p$, the maximum seasonal MODIS $f_p$, and the predicted RADARSAT-2 $f_{pk}$ for 2009-2011 is shown in Figure 8.

**Reviewer #1**
190 – "but" is an extra word here.

**Howell et al.**
Removed.

**Reviewer #1**
192 – Do you mean Figure 8 here?

**Howell et al.**
Yes. Changed to Figure 8.

**Reviewer #1**
215 – "The origin of the some of the ice" extra words here.

**Howell et al.**
Yes. Removed "the some of".

**Reviewer #1**
239 – "Overall, within the: : : ": Revisit sentence structure here.

**Howell et al.**
Revised:
Overall, within the Viscount-Melville Sound region of CAA there is a period for which a significant statistical relationship exists between RADARSAT-2 $f_{pk}$ and the summer ice area before sea ice dynamics degrades the relationship.

**Reviewer #1**

253 – "Was found to be excellent agreement": Missing "in" here.

**Howell et al.**

Added "in".

**Reviewer #1**

249 – "maybe" should be "may be" in this context.

**Howell et al.**

Changed.

---

## Author Response (AR1)

Dear Dr. Howell, dear co-authors,

I note that all three reviewers had comments regarding the results presented in figure 7. Although in your responses you have already added some material (panel b), I suggest that you consider the opportunity to add add more quantitative information about the lack of variability in the pixel surrounding the LiDAR site (either using a 3x3 window as suggested specifically by reviewer 3, or something that you think would be more appropriated).

At this stage, please, submit your revised manuscript with the edits you have shown in your responses.

**Regards**

**Howell et al.**

We have addressed all the concerns of the Reviewers and the result is a much-improved manuscript.

With regard to your comment we think we have already addressed that point about adding additional quantitative information about the lack of variability surrounding the LiDAR site. Specifically, we added aerial photography acquired over and beside the LiDAR site in 2012 (see Figure below from Scharien et al. (2014) that shows the aerial photograph coverage) and then compared those pond fraction estimates with the RADARSAT-2 peak melt pond fraction in 2012 when there was discrepancy between the LiDAR and the RADARSAT-2 pond fraction estimate. This comparison represents a wide area with about 861 samples. Figure 7b represents a distribution plot of this comparison showing how the melt pond fraction spatial variability surrounding the LiDAR site. We think that perhaps we were not very clear in the text describing this additional comparison and have revised the text in Section 3.2 as follows:

To give spatial context beyond the single point comparison at the LiDAR site, Figure 7b shows the distribution of RADARSAT-2  $f_{pk}$  and the  $f_p$  determined from aerial photo observations on June 22nd, 2012 near Resolute. The aerial photographs were acquired within 1 week of  $f_{pk}$  coverage being observed at the LiDAR site. The comparison was done by averaging all RADARSAT-2 pixels within each aerial photo (123 photos) which represents ~861 samples. The mean aerial photograph  $f_p$  was 0.54 and RADARSAT-2  $f_{pk}$  was 0.53 with an the RMSE of 0.10 and bias of 0. The distributions are in reasonably good agreement but RADARSAT-2 values are slightly narrower than the distribution of  $f_p$  from the aerial photographs. It is likely the RADARSAT-2 distribution is narrow on the left tail because our method captures peak pond coverage and some of the regions photographed were before or after their seasonal peak. We attribute the narrow right tail to the documented underestimation of equation (1) from Scharien et al. (2017). However, it is notable that both RADARSAT-2 and the aerial photograph datasets capture the same bimodal  $f_p$  distribution, with the first mode around 0.4-0.5 characterizing rougher sea ice areas and the second mode around 0.7 capturing smooth flooded sea ice. Location of aerial photography:

Figure 1. Map showing location of study area adjacent to the hamlet of Resolute Bay, NU, in the central Canadian Arctic Archipelago. Aerial photography flight lines over Parry and Field sites are shown along with outlines of 75 km × 25 km (Parry) and 25 km × 25 km (Field) Radarsat-2 scenes. The shaded region over Parry denotes the overlapping portion of scenes acquired over the site.

**Reviewer #1**

**Received and published: 10 August 2020**

This manuscript uses RADARSAT-2 imagery to derive peak melt pond fraction values for sea ice in the Canadian Arctic Archipelago between 2009 and 2018. The basic method for deriving peak pond fraction was developed in an earlier publication, and this work applies that method to a larger dataset from a different satellite. The manuscript is well written and has only a few grammatical errors that are noted below. The results presented offer valuable insight into sea ice trends and variability in the CAA. However, there are a few issues with the validation of the RADARSAT-2 derived data that should be fixed or clarified prior to publication.

**Howell et al.**

We thank this reviewer for her/his comments that have improved this manuscript considerably. We have incorporated almost all of her/his suggestions.

**Reviewer #1**

General Comments

You define  $f_p$  as melt pond fraction. Throughout the paper you also use  $f_p$  to refer to peak melt pond fraction calculated from RADARSAT-2. It would improve clarity to separate the notation for these two different parameters.

**Howell et al.**

Very good suggestion. We have chosen to define peak melt pond fraction as  $f_{pk}$  and have changed the text throughout the manuscript to reflect this new notation.

**Reviewer #1**

There are two issues with the in-situ comparison:

1. The spatial footprint of the LIDAR scans from Landy et al., (2014) are small in comparison to the 100m resolution of RADARSAT-2 data used. These in-situ datasets would only cover 1-2 pixels in the radar image. Does this area represent the whole region? Perovich (2002) determined the aggregate scale (area at which a sample can be considered representative of the larger region) at SHEBA to be multiple kilometers. If the aggregate scale is much lower in the CAA (more homogeneous ice cover) this should be discussed.

**Howell et al.**

It is true the LiDAR areas would cover only  $\sim$ 1-2 pixels, however we only compared the LiDAR pond fraction to the  $\sim$ 1-2 RADARSAT-2 pixels directly coincident with the site. Therefore, we are not validating RADARSAT-2 melt pond fraction against a spot LiDAR *in situ* measurements, we are just validating the entire 100 m LiDAR melt pond fraction directly at the sampling site. In this case, it does not matter whether the *in situ* samples are representative of the aggregate scale. We have clarified this in text so other readers to confuse other readers:

Revised Section 3.2

Figure 7a compares the time series of the entire 100 m LiDAR  $f_k$  coincident with the  $f_{pk}$  determined from RADARSAT-2 at the coinciding pixels.

**Reviewer #1**

2. Two in-situ samples are not enough to assess the accuracy of this method given the error presented in Figure 7. Here the prediction for 2011 is correct and the prediction for 2012 is not. On line 180 you state that the error is 0.1, but it looks more like 0.2 in the figure. Have you considered other in-situ datasets? For example, the three years of melt pond fraction timeseries observed on landfast ice near Utqiagvik, AK described in Polashenski et al., (2012)?

**Howell et al.**

We are limited by the scarcity of *in situ* melt pond fraction observations in the CAA and would have used more if we could. Moreover, finding observations that coincide with peak pond fraction further adds to the scarcity problem and the MODIS analysis was attempt to alleviate this problem. Unfortunately, we cannot use the *in situ* melt pond fraction dataset from Polashenski et al. (2012) because our RADARSAT-2 data only has consistent coverage in the Canadian Arctic waters in accordance with the operational domain of the Canadian Ice Service and therefore the Chukchi Sea is not covered. Despite having only two *in situ* samples, they least cover a long temporal time period allowing us to test whether RADARSAT-2 picks out the seasonal mean pond fraction obtained over and adjacent the LiDAR site in 2012 from Scharien et al. (2014), which we have made use of to compare with RADARSAT-2  $f_{pk}$  estimates.

We have added a new Figure 7b with the aerial photograph data and revised the following sections:

**Revised Section 3.2**

Figure 7a compares the time series of the entire 100 m LiDAR melt pond fraction coincident with the  $f_{pk}$  determined from RADARSAT-2 at the coinciding pixels. For 2011, RADARSAT-2  $f_{pk}$ corresponds to the end of stage I and beginning of stage II thus providing a very good representation of the seasonal peak of the  $f_p$ , when the melt pond control on heat uptake and ice decay, through the icealbedo feedback, is greatest. For 2012, RADARSAT-2  $f_{pk}$  also corresponds to the end of stage I and beginning of stage II but is ~0.20 lower than *in situ*  $f_p$  values. This is likely due to the short duration but very high maximum  $f_p$  of 0.78 in 2012 as Scharien et al. (2017) found that equation (1) sometimes underestimates very high  $f_p$  due to the low  $\gamma^{\rho}$  signal associated with very smooth FYI.

Figure 7b shows the distribution of RADARSAT-2  $f_{pk}$  and the  $f_p$  determined from aerial photo observations on June 22nd, 2012 near Resolute. The aerial photographs were acquired within 1 week of  $f_{pk}$  coverage being observed at the LiDAR site. The comparison was done by averaging all RADARSAT-2 pixels within each aerial photo. The mean aerial photograph  $f_p$  was 0.54 and RADARSAT-2  $f_{pk}$  was 0.53 with an the RMSE of 0.10 and bias of 0. The distributions are in reasonably good agreement but RADARSAT-2 values are slightly narrower than the distribution of  $f_p$  from the aerial photographs. It is likely the RADARSAT-2 distribution is narrow on the left tail because our method captures peak pond coverage and some of the regions photographed were before or after their seasonal peak. We attribute the narrow right tail to the documented underestimation of equation (1) from Scharien et al. (2017). However, it is notable that both RADARSAT-2 and the aerial photograph datasets capture the same bimodal  $f_p$  distribution, with the first mode around 0.4-0.5 characterizing rougher sea ice areas and the second mode around 0.7 capturing smooth flooded sea ice.

**Figure 7**. a) Temporal evolution of observed melt pond fraction ( $f_p$ ) and RADARSAT-2 peak melt pond fraction ( $f_{pk}$ ) at *in situ* observations sites for 2011 (74.7229°N; 95.1763°W) and 2012 (74.7264°N; 95.5772°W). b) Frequency distribution of RADARSAT-2  $f_{pk}$  and aerial photograph  $f_p$  observations in Resolute Passage on June 22, 2012; the pink vertical link represents the mean LiDAR  $f_p$  on June 22, 2012.

**Revised Section 2.1**

Aerial photographs of estimated  $f_p$  directly over the LiDAR site and the adjacent sea ice area away from land and open water were also obtained on June 22, 2012. The aerial photographs have a pixel resolution 0.22 m resolution, cover 750 m by 750 m. In total, 123 aerial photographs of  $f_p$  were used and a complete description of the dataset is provided in Scharien et al. (2014).

Added Reference

Scharien, R. K., Hochheim, K., Landy, J., and Barber, D. G.: First-year sea ice melt pond fraction estimation from dual-polarisation C-band SAR – Part 2: Scaling in situ to Radarsat-2, The Cryosphere, 8, 2163–2176, https://doi.org/10.5194/tc-8-2163-2014, 2014.

**Reviewer #1**

Lines 183-194: What is the conclusion from the comparisons with MODIS? You note the reasons why RADARSAT-2 derived  $f_p$  and MODIS  $f_p$  could be misaligned (i.e. that the MODIS product is an 8-day average and peak ponding occurs on short timescales), and I am left with the impression that the MODIS data do not agree with your results. I would suggest expanding or clarifying the statistical analysis here. In Figure 8, both 2010 and 2011 make the RADARSAT-2 look statistically different than MODIS. The mean (blue line) of RADARSAT-2 is approximately equal to the max (top whisker) of MODIS.

**Howell et al.**

This is a good point raised by the Reviewer and we were not definitive in our wording based on the boxplots. The conclusion is that RADARSAT-2 pond fraction is higher on average than MODIS because the MODIS 8-day product is not representative of  $f_{pk}$  in the CAA. The weekly boxplots and max MODIS pond fraction boxplot all support this conclusion. We note that the box plot of maximum  $f_p$  from MODIS does capture some regions at peak during the 8-day time series. Another point is that MODIS estimation error needs to be acknowledged because although it is treated here as validation for the RADARSAT-2  $f_{pk}$  estimates and rightly so but it has its own error component. We clarify this section in here as follows:

**Revised Section 3.2**

The seasonal time series of the 8-day composite MODIS  $f_p$ , the maximum seasonal MODIS  $f_p$ and the predicted RADARSAT-2  $f_{pk}$  for 2009-2011 is shown in Figure 8. MODIS  $f_p$  observations within the CAA indicate initial pond formation occurred in May for all years with  $f_{pk}$  reached in mid-July for 2009 and in early June for 2010 and 2011. Compared to the RADARSAT-2  $f_{pk}$  values, the peak MODIS  $f_p$  is ~0.20 smaller. RADARSAT-2  $f_{pk}$  is higher on average than MODIS because the MODIS 8-day product does not represent  $f_{pk}$ . The MODIS  $f_p$  observations are determined weekly using 8-day composite image products that would include some melt pond formation and drainage processes priorto, and after, the seasonal peak. Moreover, MODIS  $f_p$  observations give the time series of  $f_p$  therefore even the highest seasonal estimated MODIS  $f_p$  is reduced because while some regions of the CAA are at their seasonal peak but others are behind or ahead. To that end, we also calculated the maximum  $f_p$  from MODIS regardless of timing during the melt season, for each pixel, also shown in Figure 8. These values more closely compare with the RADARSAT-2  $f_{pk}$  but are still ~0.05 smaller on average. Even the maximum  $f_p$  from MODIS is from an 8-day running mean of daily pond fraction estimates, so will underestimate the  $f_{pk}$  if the duration of peak ponding is

**Reviewer #1**

251 - In 214-231 you posit that the predictive power of this method only holds for landfast ice (i.e. when ice breakup is due to thermodynamics and not due to ice motion), how would this method be applicable to pan-Arctic estimates?

**Howell et al.**

In that case a Lagrangian tracking approach would be needed or the integrated melt pond fraction could be used with evolving sea ice extent. In both cases, significant testing would be required. We are working on this, but it is considerably outside the scope of this analysis.

**Reviewer #1**

Technical Corrections 59-61 – Run-on sentence.

**Howell et al.**

**Revised Introduction**

Model simulations have been utilized to understand the current and predicted future variability of sea ice conditions in the CAA (e.g. Dumas et al., 2006; Sou and Flato, 2009, Howell et al., 2016; Laliberté et al., 2016; Hu et al., 2018; Laliberté et al., 2018). However, modeling the CAA still remains challenging because complex sea ice dynamic and thermodynamic processes are often not accurately resolved in its narrow channels and inlets.

**Reviewer #1**

97 - "during April in within the CAA": Extra "in" here.

**Howell et al.**

Removed

**Reviewer #1** 152 – This sentence is unclear.**

**Howell et al.**

Revised:

What is interesting in Figure 5a is that the mean RADARSAT-2  $f_{peak}$  in 2009 was lower than all years from 2014-2018 (with the exception of 2016) despite the CAA containing less MYI area.

**Reviewer #1**

154 – "in addition" and "also" are redundant here.

**Howell et al.**

Removed "also"

**Reviewer #1**

161 - 3.2 header has extra "and". Also consider including oxford comma in this list for added clarity.

**Howell et al.**

Revised:

3.2 Comparison of RADARSAT-2  $f_{pk}$  with Sentinel-1  $f_{pk}$ , in situ  $f_p$ , and MODIS  $f_p$

**Reviewer #1**

183 – Again a stylistic choice, but I find oxford commas to be helpful for clarity.

**Howell et al.**

Revised:

The seasonal time series of the 8-day composite MODIS  $f_p$ , the maximum seasonal MODIS  $f_p$ , and the predicted RADARSAT-2  $f_{pk}$  for 2009-2011 is shown in Figure 8.

**Reviewer #1**

190 – "but" is an extra word here.

**Howell et al.**

Removed.

**Reviewer #1** 192 – Do you mean Figure 8 here?

**Howell et al.**

Yes. Changed to Figure 8.

Reviewer #1

215 – "The origin of the some of the ice" extra words here.

**Howell et al.** Yes. Removed "the some of".

**Reviewer #1**

239 – "Overall, within the: : : ": Revisit sentence structure here.

**Howell et al.**

Revised:

Overall, within the Viscount-Melville Sound region of CAA there is a period for which a significant statistical relationship exists between RADARSAT-2  $f_{pk}$  and the summer ice area before sea ice dynamics degrades the relationship.

"in".

**Reviewer #1**

253 – "Was found to be excellent agreement": Missing "in" here.

**Howell et al.**

Added

**Reviewer #1**

249 – "maybe" should be "may be" in this context.

**Howell et al.**

Changed.

**Reviewer #2**

The manuscript uses RADARSAT-2 data to estimate melt pond fraction within the Canadian Arctic. The manuscript is clear and well written with figures clearly supporting the presented results and the discussion.

**Howell et al.**

We thank this reviewer for her/his comments that have improved this manuscript. We have incorporated almost all this reviewer's suggestions.

**Reviewer #2**

I found the investigation into the correlation between the different regions and the melt pond fraction one of the most important findings of this study. Maybe this finding could be more explicitly stated in the abstract and also in the conclusion? "Static/stable sea ice regions showed a higher detrended correlation." The mentioning of several regions is a bit vague.

**Howell et al.**

Agreed.

**Revised Abstract:**

Dynamically stable sea ice regions within the CAA exhibited higher detrended correlations between RADARSAT-2  $f_{pk}$  summer sea ice area.

**Revised Conclusions:**

The results presented in this study indicate that dynamically stable sea ice regions within the CAA exhibit a higher detrended correlation between RADARSAT-2  $f_{pk}$  and summer sea ice area.

**Reviewer #2**

Single pol RADARSAT-2 data was used, why is that? Was the combination of HH + HV lacking? Or did the HH-channel contribute sufficient information? This may have been covered in earlier work by e.g. Scharien et al., but would then be worth reiterating.

**Howell et al.**

Single pol RADARAT-2 was used for two reasons. The first is that Scharien et al. (2017) found the HV data produced noisy results and the second there is not sufficient HV imagery in the early of the RADARSAT-2 to cover CAA. The latter is because only in the recent years has HH+HV been ordered operationally throughout the CAA.

**Revised Data:**

We limited our analysis to only RADARAT-2 images at HH polarization because Scharien et al. (2017) found HV produced noisy results in addition to there not being sufficient imagery at HV polarization in the early period of the RADARSAT-2 record to cover CAA in April.

**Reviewer #2**

The in-situ area only covers areas with a relatively high proportion of melt ponds, were any other in-situ data available that could be used for the validation with a smaller proportion of melt ponds? Moreover, the area covered for the in-situ data is rather small compared to the pixel size of the RADARSAT-2 images. Are there larger datasets, either more locations or covering a larger area that could be used to strengthen the argument?

**Howell et al.**

Yes, we do have aerial photograph estimates of melt pond fraction obtained over and adjacent to the LiDAR site in 2012 from Scharien et al. (2014), which we have made use of to compare with RADARSAT-2  $f_{pk}$  estimates. We have added a new Figure 7b with the aerial photograph data and revised the following sections:

**Revised Section 3.2**

Figure 7a compares the time series of the entire 100 m LiDAR melt pond fraction coincident with the  $f_{pk}$  determined from RADARSAT-2 at the coinciding pixels. For 2011, RADARSAT-2  $f_{pk}$ corresponds to the end of stage I and beginning of stage II thus providing a very good representation of the seasonal peak of the  $f_p$ , when the melt pond control on heat uptake and ice decay, through the icealbedo feedback, is greatest. For 2012, RADARSAT-2  $f_{pk}$  also corresponds to the end of stage I and beginning of stage II but is ~0.2 lower than *in situ*  $f_p$  values. This is likely due to the short duration but very high maximum  $f_p$  of 0.78 in 2012 as Scharien et al. (2017) found that equation (1) sometimes underestimates very high  $f_p$  due to the low  $\gamma^{\rho}$  signal associated with very smooth FYI.

Figure 7b shows the distribution of RADARSAT-2  $f_{pk}$  and the  $f_p$  determined from aerial photo observations on June 22nd, 2012 near Resolute. The aerial photographs were acquired within 1 week of  $f_{pk}$  coverage being observed at the LiDAR site. The comparison was done by averaging all RADARSAT-2 pixels within each aerial photo. The mean aerial photograph  $f_p$  was 0.54 and RADARSAT-2  $f_{pk}$  was 0.53 with an the RMSE of 0.10 and bias of 0. The distributions are in reasonably good agreement but RADARSAT-2 values are slightly narrower than the distribution of  $f_p$  from the aerial photographs. It is likely the RADARSAT-2 distribution is narrow on the left tail because our method captures peak pond coverage and some of the regions photographed were before or after their seasonal peak. We attribute the narrow right tail to the documented underestimation of equation (1) from Scharien et al. (2017). However, it is notable that both RADARSAT-2 and the aerial photograph datasets capture the same bimodal  $f_p$  distribution, with the first mode around 0.4-0.5 characterizing rougher sea ice areas and the second mode around 0.7 capturing smooth flooded sea ice.